# Causal associations between risk factors and common diseases inferred from GWAS summary data

Zhihong Zhu[1], Zhili Zheng[1,2], Futao Zhang[1], Yang Wu[1], Maciej Trzaskowski[1], Robert Maier [1], Matthew R. Robinson [1], John J. McGrath [3,4,5], Peter M. Visscher [1,3], Naomi R. Wray [1,3] & Jian Yang [1,3]

Health risk factors such as body mass index (BMI) and serum cholesterol are associated with many common diseases. It often remains unclear whether the risk factors are cause or consequence of disease, or whether the associations are the result of confounding. We develop and apply a method (called GSMR) that performs a multi-SNP Mendelian randomization analysis using summary-level data from genome-wide association studies to test the causal associations of BMI, waist-to-hip ratio, serum cholesterols, blood pressures, height, and years of schooling (EduYears) with common diseases (sample sizes of up to 405,072). We identify a number of causal associations including a protective effect of LDL-cholesterol against type-2 diabetes (T2D) that might explain the side effects of statins on T2D, a protective effect of EduYears against Alzheimer's disease, and bidirectional associations with opposite effects (e.g., higher BMI increases the risk of T2D but the effect of T2D on BMI is negative).

[1] Institute for Molecular Bioscience, The University of Queensland, Brisbane, QLD 4072, Australia. [2] The Eye Hospital, School of Ophthalmology & Optometry, Wenzhou Medical University, Wenzhou, 325027 Zhejiang, China. [3] Queensland Brain Institute, The University of Queensland, Brisbane, QLD 4072, Australia. [4] Queensland Centre for Mental Health Research, The Park Centre for Mental Health, Wacol, QLD 4072, Australia. [5] National Centre for Register-Based Research, Aarhus University, 8000 Aarhus C, Denmark. Correspondence and requests for materials should be addressed to J.Y. (email: jian.yang@uq.edu.au)

Health risk factors such as body mass index (BMI), serum cholesterol, and blood pressure are associated with many human common diseases[1,2], e.g., being overweight is associated with increased risk to cardiovascular diseases (CVD)[3] and type-2 diabetes (T2D)[4]. These associations are usually derived from observational studies that cannot distinguish whether the risk factors are "upstream" causal factors, "downstream" consequences of the diseases, or confounding factors associated with both the exposures and outcomes. The randomized controlled trial (RCT) is considered to be the gold standard approach to test for causality. For instance, LDL-cholesterol (LDL-c) was initially found to be associated with coronary artery disease (CAD) in an observational study[5], and the association was subsequently confirmed to be causal by RCTs[6,7]. However, RCTs are time-consuming, expensive, and sometimes impractical or even unethical[8]. It is not feasible to design RCTs that can test many different interventions simultaneously. Genetic methods are useful to infer causality because genetic variants are present from birth and therefore unlikely to be confounded with environmental factors. Mendelian randomization (MR) is an analysis that uses genetic variants, which are expected to be independent of confounding factors, as instrumental variables to test for causality[9–11]. MR can be used to infer credible causal associations when RCTs are not feasible or as a strategy to rank order candidate causal associations to be prioritized for follow-up in RCTs. MR is becoming increasingly efficient and cost-effective given the ever-growing data curated from recent genome-wide association studies (GWAS). The large amount of GWAS data available in the public domain provide a great opportunity for methods that are able to make inference about causality by integrating summary-level GWAS data from different studies[12–16]. We have previously shown that the power of an MR analysis could be greatly improved by exploiting GWAS summary data from two independent studies with large sample sizes, and have applied a summary data-based MR (SMR) approach to test whether the effects of genetic variants on a phenotype are mediated by gene expression[17].

In this study, we extend the SMR approach to a more general form (generalized SMR or GSMR) by leveraging power from multiple genetic variants accounting for linkage disequilibrium (LD) between the variants, and demonstrate by simulation that GSMR is more powerful than existing summary data-based MR methods[12,13,18]. Separation of signals of causality from pleiotropy (a single locus directly affecting multiple phenotypes, also called type-II pleiotropy[19]) and further separation of marginal effect from conditional effect (the net effect of a risk factor on the outcome accounting for the effects of other risk factors, e.g., there is no effect of HDL cholesterol on CAD correcting for the other serum cholesterol levels[20,21]) are recognized issues that require careful interpretation in MR analyses[19]. We develop a method (HEIDI-outlier) to detect and eliminate genetic instruments that have apparent pleiotropic effects on both risk factor and disease, and another method (multi-trait-based conditional and joint analysis, or mtCOJO) to estimate the effect of a risk factor on disease conditioning on other risk factors. All methods developed in this study only require summary-level data (with LD between genetic variants from a reference sample with individual-level data), providing a great flexibility to integrate data from multiple studies. We apply the methods to publicly available data of very large sample sizes (n = up to 405,072 for risk factors and 184,305 for diseases) to test the causal associations between health risk factors such as BMI, serum cholesterol levels and blood pressure levels and a range of human common diseases. Our study develops powerful tools to integrate summary data from large studies to infer causality, and provides important candidates to be prioritized for further studies in medical research and for drug discovery.

## Results

**Overview of the methods.** Let $y$ be the liability of a disease on the logit scale, $x$ be a risk factor in standard deviation (SD) units and $z$ be the genotype of a SNP (coded as 0, 1, or 2). The MR estimate of the causal effect of risk factor on disease[9] is $\hat{b}_{xy} = \hat{b}_{zy}/\hat{b}_{zx}$, where $b_{zy}$ is the effect of $z$ on $y$ on the logit scale (logarithm of odds ratio, logOR), $b_{zx}$ is the effect of $z$ on $x$, and $b_{xy}$ is the effect of $x$ on $y$ free of confounding from non-genetic factors (note that $b_{xy}$ can be approximately interpreted as logOR; see below). SMR is a flexible and powerful MR approach that is able to estimate and test the significance of $b_{xy}$ using the estimates of $b_{zx}$ and $b_{zy}$ from independent samples[17]. If there are multiple independent (or nearly independent) SNPs associated with $x$ and the effect of $x$ on $y$ is causal, then all the $x$-associated SNPs will have an effect on $y$ through $x$ (Fig. 1a). In this case, $b_{xy}$ at any of the $x$-associated SNPs is expected to be identical in the absence of pleiotropy[13,16,22] as all the SNP effects on $y$ are mediated by $x$ (Fig. 1b). Therefore, increased statistical power can be achieved by integrating the estimates of $b_{xy}$ from all the $x$-associated SNPs

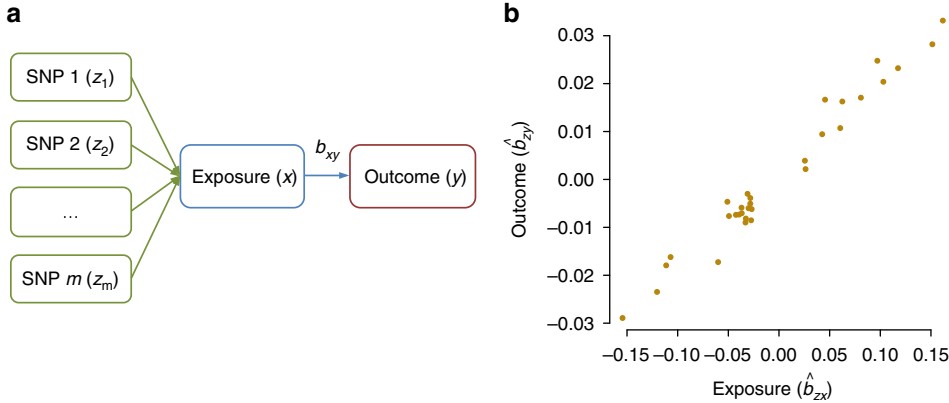

**Fig. 1** Leveraging multiple independent genetic instruments ($z$) to test for causality. Shown in panel **a** is a schematic example that if an exposure ($x$) has an effect on an outcome ($y$), any instruments (SNPs) causally associated with $x$ will have an effect on $y$, and the effect of $x$ on $y$ ($b_{xy}$) at any of the SNPs is expected to be identical. This is further illustrated in a toy example in panel **b** that under a causal model, for the SNPs associated with $x$, the estimated effect of $z$ on $y$ ($\hat{b}_{zy}$) should be linearly proportional to the estimated effect of $z$ on $x$ ($\hat{b}_{zx}$) and the ratio between the two is an estimate of the mediation effect of $x$ on $y$, i.e., $\hat{b}_{xy} = \hat{b}_{zy}/\hat{b}_{zx}$

using a generalized least squares (GLS) approach (Methods). The GSMR method essentially implements SMR analysis for each SNP instrument individually, and then integrates the $b_{xy}$ estimates of all the SNP instruments by GLS, accounting for the sampling variance in both $\hat{b}_{zx}$ and $\hat{b}_{zy}$ for each SNP and the LD among SNPs. It is important to note that in accordance with one of the basic assumptions for MR[9], only the SNPs that are strongly associated with the risk factor should be used as the instruments for MR analyses including GSMR. We demonstrate using simulations (Supplementary Note 1) that if we use independent SNPs that are associated with the exposure at $P < 5 \times 10^{-8}$, there is no inflation in the GSMR test-statistics under the null hypothesis that $b_{xy} = 0$ (Supplementary Fig. 1a), that the estimate of $b_{xy}$ by GSMR is unbiased under the alternative hypothesis that $b_{xy} \neq 0$ (Supplementary Table 1), and that $b_{xy}$ approximately equals to logOR (where OR is the effect of risk factor on disease in observational study without confounding) (Supplementary Fig. 2). GSMR accounts for LD if the SNP instruments are not fully independent. This is demonstrated by the simulation that in the presence of LD the test-statistic is well calibrated under the null (Supplementary Fig. 1b) and that the estimate of $b_{xy}$ is unbiased under the alternative (Supplementary Table 1). In comparison with the existing methods that use summary data to make causal inference[12,13,16,18], GSMR is more powerful as demonstrated by simulation (Supplementary Fig. 3) because GSMR accounts for the sampling variance in both $\hat{b}_{zx}$ and $\hat{b}_{zy}$ while the other approaches assume that $b_{zx}$ is estimated without error.

Pleiotropy is an important potential confounding factor that could bias the estimate and often results in an inflated test-statistic in a MR analysis[9,10,13,19]. We propose a method (called HEIDI-outlier) to detect pleiotropic SNPs at which the estimates of $b_{xy}$ are significantly different from expected under a causal model, and remove them from the GSMR analysis (Methods). The power of detecting a pleiotropic SNP depends on the sample sizes of the GWAS data sets and the deviation of $\hat{b}_{xy}$ estimated at the pleiotropic SNP from the causal model. We have demonstrated by simulation based on a causal model with pleiotropy that the power of HEIDI-outlier is high especially when the pleiotropic effects are large (Supplementary Fig. 4a). There are certainly pleiotropic outliers (e.g., those with very small effects) not detected by HEIDI-outlier. Nevertheless, these undetected pleiotropic effects do not seem to bias the GSMR estimate (Supplementary Fig. 4b), in contrast to a small bias in the estimate from Egger regression (MR-Egger) which is thought to be free of confounding from pleiotropy[13]. Our simulation results also show that the GSMR estimate of $b_{xy}$ is not significantly different from zero under a pleiotropic model without causal effect in the presence or absence of LD (Supplementary Table 2).

We further develop an approximate method (called mtCOJO; URLs) that only requires summary data to conduct a GWAS analysis for a phenotype conditional on multiple covariate phenotypes (Methods). The purpose of developing this method is to estimate the effect of a risk factor on disease adjusting for other risk factors (Methods; Supplementary Note 2; Supplementary Fig. 5), which helps to infer whether the marginal effect of the risk factor on disease depends on other risk factors, and to predict the joint effect of multiple risk factors on disease. It is of note that mtCOJO is free of bias due to shared environmental or genetic effect between the phenotype and covariate as described in Aschard et al.[23] (Supplementary Fig. 6).

**The effects of seven health risk factors on common diseases.** We applied the methods to test for causal associations between seven health risk factors and common diseases using data from multiple large studies. The risk factors are BMI, waist-to-hip ratio adjusted for BMI (WHRadjBMI), HDL cholesterol (HDL-c), LDL-c, triglycerides (TG), systolic blood pressure (SBP), and diastolic blood pressure (DBP). We chose these risk factors because of the availability of summary-level GWAS data from large samples ($n = 108,039$–$322,154$) (Supplementary Table 3). We accessed data for BMI, WHRadjBMI, HDL-c, LDL-c and TG from published GWAS[24–26], and data for SBP and DBP from the subgroup of UK Biobank (UKB)[27] with genotyped data released in 2015. We selected SNPs at a genome-wide significance level ($P_{GWAS} < 5 \times 10^{-8}$) using the clumping algorithm ($r^2$ threshold = 0.05 and window size = 1 Mb) implemented in PLINK[28] (Methods). Note that the GSMR method accounts for the remaining LD not removed by the clumping analysis. There were $m = 84$, 43, 159, 141, 101, 28, and 29 SNPs for BMI, WHRadjBMI, HDL-c, LDL-c, TG, SBP and DBP, respectively, after clumping. These SNP instruments are nearly independent as demonstrated by the distribution of LD scores computed from the instruments for each trait (Supplementary Fig. 7). We only included in the analysis the near-independent SNPs for the ease of directly comparing the results from GSMR with those from other methods that do not account for LD (e.g., MR-Egger). Our simulation result suggests that the gain of power by including SNPs in LD is limited (Supplementary Fig. 8). Moreover, although the GSMR approach accounts for LD, including many SNPs in moderate to high LD often results in the **V** matrix being non-invertible (Methods).

The summary-level GWAS data for the diseases were computed from two independent community-based studies with individual-level SNP genotypes, i.e., the Genetic Epidemiology Research on Adult Health and Aging[29] (GERA) ($n = 53,991$) and the subgroup of UKB[27] ($n = 108,039$). We included in the analysis 22 common diseases as defined in the GERA data, and added an additional phenotype related to comorbidity by counting the number of diseases affecting each individual (i.e., disease count) as a crude index to measure the general health status of an individual (Supplementary Table 4). We performed genome-wide association analyses of the 23 disease phenotypes in GERA and UKB separately (Methods). We assessed the genetic heterogeneity of a disease between the two cohorts by a genetic correlation ($r_g$) analysis using the bivariate LD score regression (LDSC) approach[30]. The estimates of $r_g$ across all diseases varied from 0.75 to 0.99 with a mean of 0.91 (Supplementary Table 4), suggesting strong genetic overlaps for the diseases between the two cohorts. We therefore meta-analyzed the data of the two cohorts to maximize power using the inverse-variance meta-analysis approach[31]. Because OR is free of the ascertainment bias in a case–control study, the effect size (logOR) of a SNP on disease in the general population can be approximated by that from a case–control study assuming that disease in the case–control study is defined similarly as that in the general population. Therefore, GSMR can be applied to data with SNP effects on the risk factor from a population-based study and SNP effects on the disease from an ascertained case–control study, and the estimated causative effect of risk factor on disease should be interpreted as that in the general population. We therefore included in the analysis summary data for 11 diseases from published case–control studies ($n = 18,759$–$184,305$) (Supplementary Table 5). The estimated SNP effects and standard errors (SE) for age-related macular degeneration (AMD) were not available in the summary data[32], which were estimated from $z$-statistics using an approximate approach (Supplementary Note 3).

We applied the HEIDI-outlier approach to remove SNPs that showed pleiotropic effects on both risk factor and disease, significantly deviated from a causal model (Methods). The LD correlations between pairwise SNPs were estimated from the

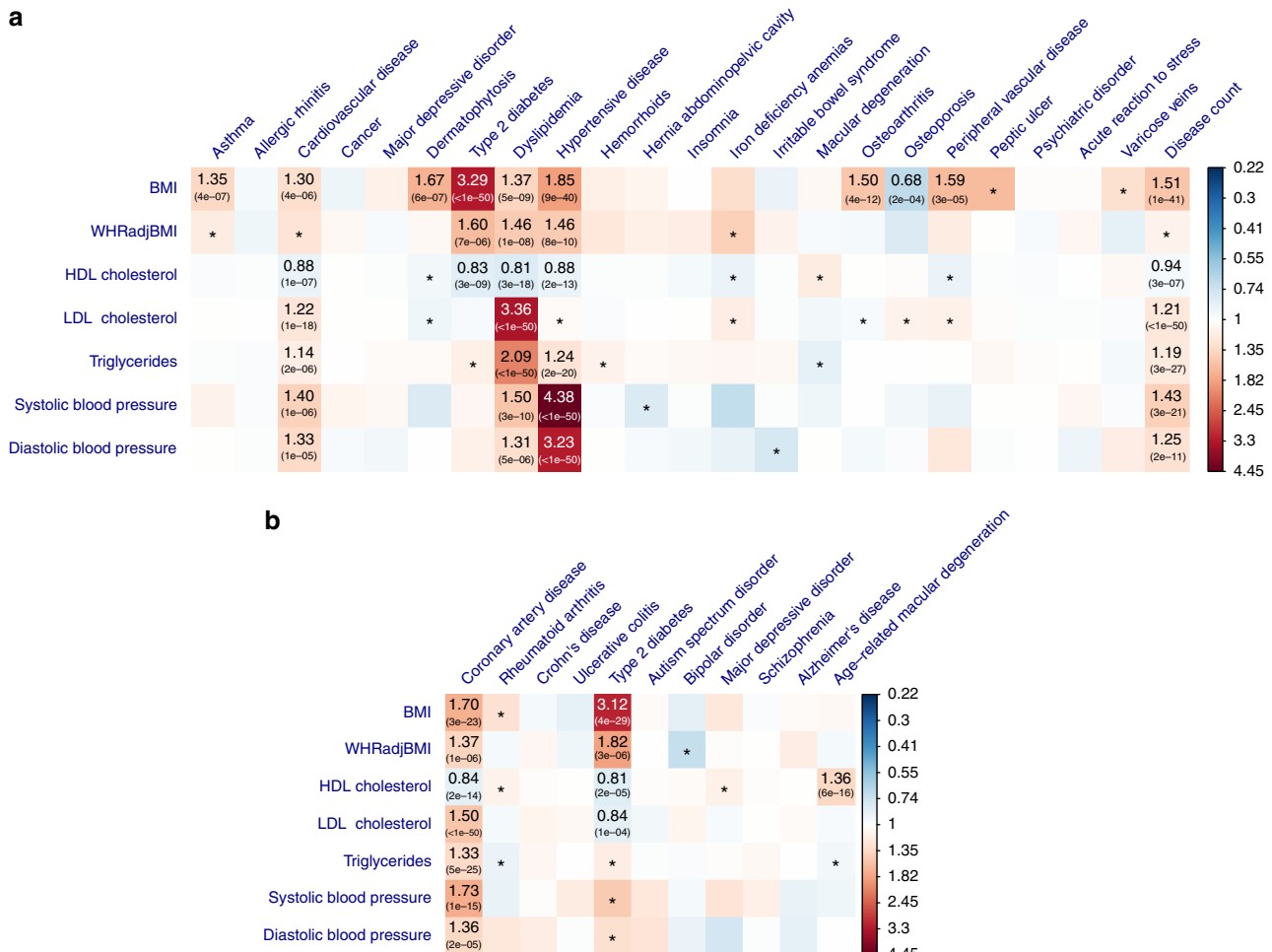

**Fig. 2** Putative causal associations between seven modifiable risk factors and common diseases. Shown are the results from GSMR analyses with disease data **a** from a meta-analysis of two community-based studies (GERA and UKB) and **b** from published independent case-control studies. Colors represent the effect sizes (as measured by odds ratios, ORs) of risk factors on diseases, red for risk effects and blue for protective effects. The significant effects after correcting for 231 tests ($P_{GSMR} < 2.2 \times 10^{-4}$) are labeled with ORs (P-values). The nominally significant effects ($P_{GSMR} < 0.05$) are labeled with "*"

Atherosclerosis Risk in Communities (ARIC) data[33] ($n = 7703$ unrelated individuals) imputed to 1000 Genomes (1000G)[34]. Using the large data sets described above, we identified from GSMR analyses 45 significant causative associations between risk factors and diseases (Supplementary Data 1; Fig. 2). We controlled the family-wise error rate (FWER) at 0.05 by Bonferroni correction for 231 tests ($P_{GSMR}$ threshold = $2.2 \times 10^{-4}$). For method comparison, we have also performed the analyses with MR-Egger[13] and the methods in Pickrell et al.[16] (Supplementary Data 2).

**Obesity and common diseases.** Results from analyses of the community-based data showed that BMI had risk effects on T2D (odds ratio, OR = 3.29), hypertensive disease (OR = 1.85), dermatophytosis (i.e., tinea) (OR = 1.67), peripheral vascular diseases (PVD) (OR = 1.59), osteoarthritis (OR = 1.50), dyslipidemia (OR = 1.37), asthma (OR = 1.35), and CVD (OR = 1.30). The risk effects of BMI on T2D, CVD, and hypertensive disease have been confirmed by RCT[35] (Supplementary Data 1), providing proof-of-principle validation. The interpretation of $OR_{(BMI \rightarrow T2D)} = 3.29$ is that people whose BMI are 1 SD (SD = 3.98 for BMI in European men corresponding to ~12 kg of weight for men of 175 cm stature; see Supplementary Table 6 for the SD of the risk factors) above the population mean will have 3.29 times increase in risk to

T2D compared with the population prevalence (~8% in the US). It is interesting to note that the estimate of $b_{xy}$ at the TCF7L2 locus strongly deviated from those at the other loci (Fig. 3), suggesting that the TCF7L2 SNP has pleiotropic effects on BMI and T2D. The TCF7L2 SNP was detected as an outlier by the HEIDI-outlier method and removed from the GSMR analysis. In addition, the risk effect of BMI on asthma is in line with the result from a recent MR study (using a weighted genetic allele score as the instrument) that higher BMI increases the risk of childhood asthma[36]. Moreover, we identified a protective effect of BMI against osteoporosis (OR = 0.68), consistent with the observed associations in previous studies[37,38]. The estimated risk effect of BMI on T2D in the community data (OR = 3.29) was similar to that in the case-control data (OR = 3.12, Fig. 2b and Supplementary Data 1). We also observed a strong risk effect of BMI on coronary artery disease (CAD) in the case-control data (OR = 1.70), in line with the risk effect of BMI on CVD (OR = 1.30) in the community data.

Being overweight is a risk factor for general health outcomes as indicated by its risk effect on disease count ($\hat{b}_{xy} = 0.41$) in the community data. The question is then how $b_{xy}$ for disease count should be interpreted. We have shown in Supplementary Fig. 9 that the estimate of $b_{xy}$ for disease status (a dichotomous phenotype to indicate whether an individual is affected by any of the 22 diseases) was very similar to that for disease count.

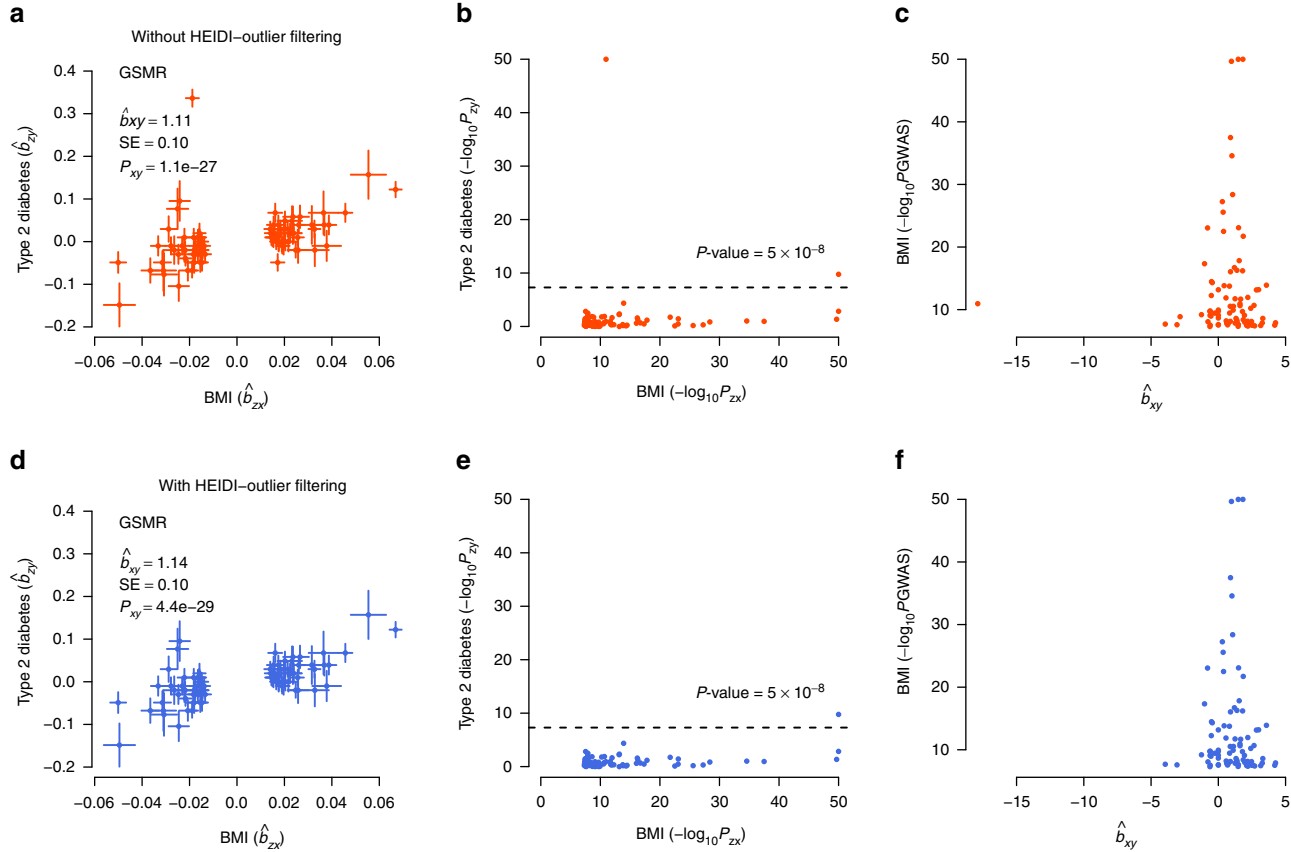

**Fig. 3** GSMR analysis to test for the effect of BMI on T2D with and without filtering the pleiotropic outliers. Shown in **a** and **b** are the plots of effect sizes and association *P*-values of all the genetic instruments from GWAS for BMI vs. those for T2D. Shown in **c** is the plot of $b_{xy}$ vs. GWAS *P*-value of BMI at each genetic variant. Shown in **d**, **e**, and **f** are the plots for the instruments after the pleiotropic outliers being removed by the HEIDI-outlier approach (see Methods for details of the HEIDI-outlier approach). Error bars in **a** and **d** represent the standard errors. The dashed lines in **b** and **e** represent the GWAS threshold *P*-value of $5 \times 10^{-8}$. The coordinates in **b**, **c**, **e**, and **f** are truncated at 50 for better graphic presentation

Although disease status and disease count are two distinct phenotypes and the analysis of disease count is more powerful, for the ease of interpretation, $b_{xy}$ for disease count can be approximately interpreted as logOR for disease status. Hence, $\hat{b}_{xy} = 0.41$ for disease count is approximately equivalent to OR = 1.51 for disease status, meaning an increase of BMI by 1 SD will increase the probability of being affected by any of the 22 diseases by a factor of ~1.5. In addition, we found that the effects of WHRadjBMI and BMI on disease were largely concordant (Supplementary Fig. 10a; Supplementary Note 4).

**Serum cholesterol levels and common diseases**. LDL-c is a known causative risk factor for CAD as confirmed by RCTs[6,7]. We found that LDL-c had a significant risk effect on dyslipidemia (OR = 3.36) and CVD (OR = 1.22) in the community data, and CAD (OR = 1.50) in the case–control data (Fig. 2). TG had a significant risk effect on dyslipidemia (OR = 2.09), hypertensive disease (OR = 1.24) and CVD (OR = 1.14) in the community data, and CAD (OR = 1.33) in the case–control data (Fig. 2). The effects of TG on diseases were largely consistent with those for LDL-c (Supplementary Fig. 10b), despite the modest phenotypic correlation between the two traits ($r_p = 0.19$ in the ARIC data). Both LDL and TG had significant risk effects on disease count in the community data (Fig. 2).

There was another example where the HEIDI-outlier approach detected strong effects due to pleiotropy. The effect of LDL-c on Alzheimer's disease (AD) was highly significant without HEIDI-

outlier filtering (OR = 1.35 and $P_{\text{GSMR}} = 7.8 \times 10^{-16}$) (Fig. 4). The HEIDI-outlier analysis flagged 16 SNPs, 12 of which are located in the *APOE* gene region (LD $r^2$ among these SNPs < 0.05) and all of which had highly significant effects on both LDL-c and AD. Excluding these SNPs makes a more conservative GSMR test because if there is a true causal relationship of increased LDL-c with AD, then the GSMR test should remain significant based on evidence from other LDL-c associated SNPs. In fact, after removing the 16 pleiotropic SNPs, the estimated effect of LDL-c on AD was not significant (OR = 1.03, $P_{\text{GSMR}} = 0.47$). Nevertheless, the multiple pleiotropic signals clustered at the *APOE* locus are worth further investigation (Supplementary Fig. 11).

We identified a significant protective effect of LDL-c against T2D (OR = 0.84, $P_{\text{GSMR}} = 1.1 \times 10^{-4}$) in the case–control data, which might explain the observation from a previous study that lowering LDL-c using statin therapy is associated with a slightly increased risk of T2D[39]. The estimate was not significant in the community data (likely due to the lack of power) but in a consistent direction (OR = 0.95, $P_{\text{GSMR}} = 0.08$). Given the strong genetic correlation between the two T2D data sets ($r_g = 0.98$, SE = 0.062) as estimated by the bivariate LDSC analysis[30], we meta-analyzed the two data sets using the inverse-variance approach, and performed the GSMR analysis to re-estimate the effect of LDL-c on T2D using the T2D meta-analysis data. The effect size was highly significant (OR = 0.88, $P_{\text{GSMR}} = 3.0 \times 10^{-7}$).

The consequences of HDL-c on health outcomes are controversial[40]. Observational studies suggest that HDL-c is associated with a reduced risk to CAD[41], whereas genetic studies

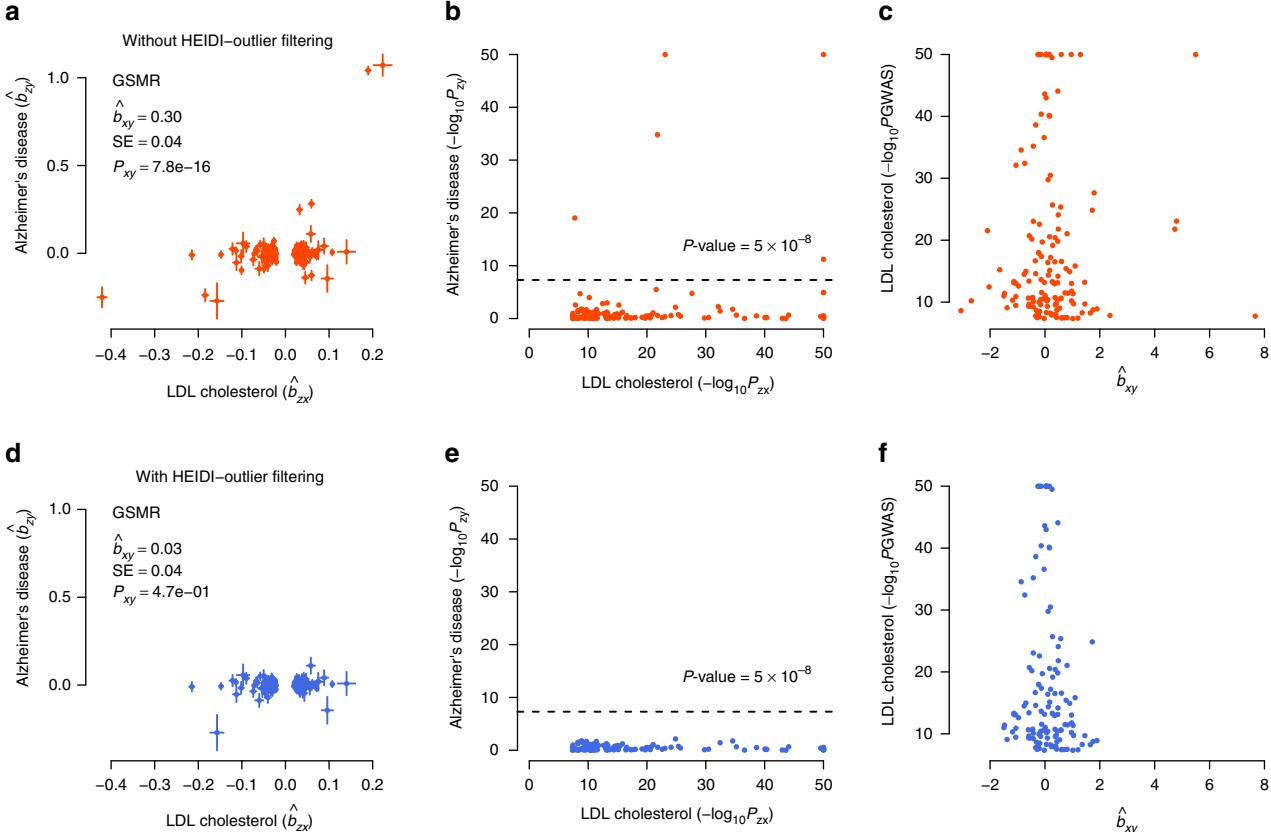

**Fig. 4** GSMR analysis to test for the effect of LDL-c on Alzheimer's disease (AD) with and without pleiotropic outliers. Shown in **a** and **b** are the plots of effect sizes and association *P*-values of the original set of instruments from GWAS for LDL-c vs. those for AD. Shown in **c** is the plot of $b_{xy}$ vs. GWAS *P*-value of LDL-c at each genetic variant. Shown in **d**, **e**, and **f** are the plots for the instruments after the pleiotropic outliers being removed by the HEIDI-outlier approach (see Methods for details of the HEIDI-outlier approach). Error bars in **a** and **d** represent the standard errors. The dashed lines in **b** and **e** represent the GWAS threshold *P*-value of $5 \times 10^{-8}$. The coordinates in **b**, **c**, **e**, and **f** are truncated at 50 for better graphic presentation

show that the effect of HDL-c on CAD is not significant conditional on LDL-c and TG[20,21]. We found that HDL-c had protective effects against T2D (OR = 0.83), hypertensive disease (OR = 0.88), CVD (OR = 0.88) and disease count (OR = 0.94) in the community data, and T2D (OR = 0.81) and CAD (OR = 0.84) in the case–control data. However, none of these effects remained significant conditioning on the other risk factors, suggesting that the marginal effects of HDL-c on diseases are dependent of the other risk factors (see below for details of the results from conditional analyses). The effect of HDL-c on dyslipidemia is negative ($b_{xy} = -0.21$ and OR = 0.81), which is obvious because one of the diagnostic criteria for dyslipidemia is an abnormally low level of HDL-c. In addition, there was a highly significant risk effect (OR = 1.36) of HDL-c on age-related macular degeneration (AMD) in the case–control data, consistent with the result from a recent MR study[42]. The associations between lipids and AMD are controversial and results from different observational studies are inconsistent[43]. Our results support the observations that increased HDL-c is associated with increased risk of AMD[43–45]. It should be noted that LDL-c and TG also appeared to be associated with AMD before HEIDI-outlier filtering but the effects were not significant after HEIDI-outlier filtering (Supplementary Fig. 12), implying that the observed association between LDL-c (or TG) and AMD in epidemiological studies[43] might be due to pleiotropy.

**Blood pressure and common diseases**. We identified significant risk effects of SBP on hypertensive disease (OR = 4.38),

dyslipidemia (OR = 1.50), CVD (OR = 1.40) and disease count (OR = 1.43) in the community data, and CAD (OR = 1.73) in the case–control data. The results for SBP and DBP were highly concordant (Fig. 2; Supplementary Fig. 10c). The risk effect of blood pressure on CAD is known to be causal as confirmed by RCTs[46,47]. Note that the power of the GSMR analysis for blood pressure was likely to be limited given the small number of instruments used ($m < 30$).

**Conditional effects of risk factors on diseases**. We have identified (from the analyses above) 45 significant causal associations between health risk factors and diseases (Fig. 2). As the risk factors are not independent, we further sought to estimate the effect of a risk factor on a disease adjusting for other risk factors. To do this, we first investigated the causal associations among the risk factors. We detected 19 significant associations by the GSMR analysis among the 7 risk factors at a FWER of 0.05 ($P_{GSMR} < 1.2 \times 10^{-3}$) (Supplementary Fig. 13). For example, BMI had a significant negative effect on HDL-c ($b_{xy} = -0.29$), and positive effects on TG ($b_{xy} = 0.28$) and DBP ($b_{xy} = 0.15$).

We developed an approach called mtCOJO (multi-trait-based conditional and joint analysis; URLs) to perform a GWAS analysis for a trait conditioning on other traits using GWAS summary data (Methods; Supplementary Fig. 5). We then re-ran the GSMR analysis using the adjusted GWAS summary data from the mtCOJO analysis (Methods). The mtCOJO analysis requires the estimates of $b_{xy}$ of the covariate risk factors on the target risk factor and disease, $r_g$ among the covariate risk factors, SNP-based

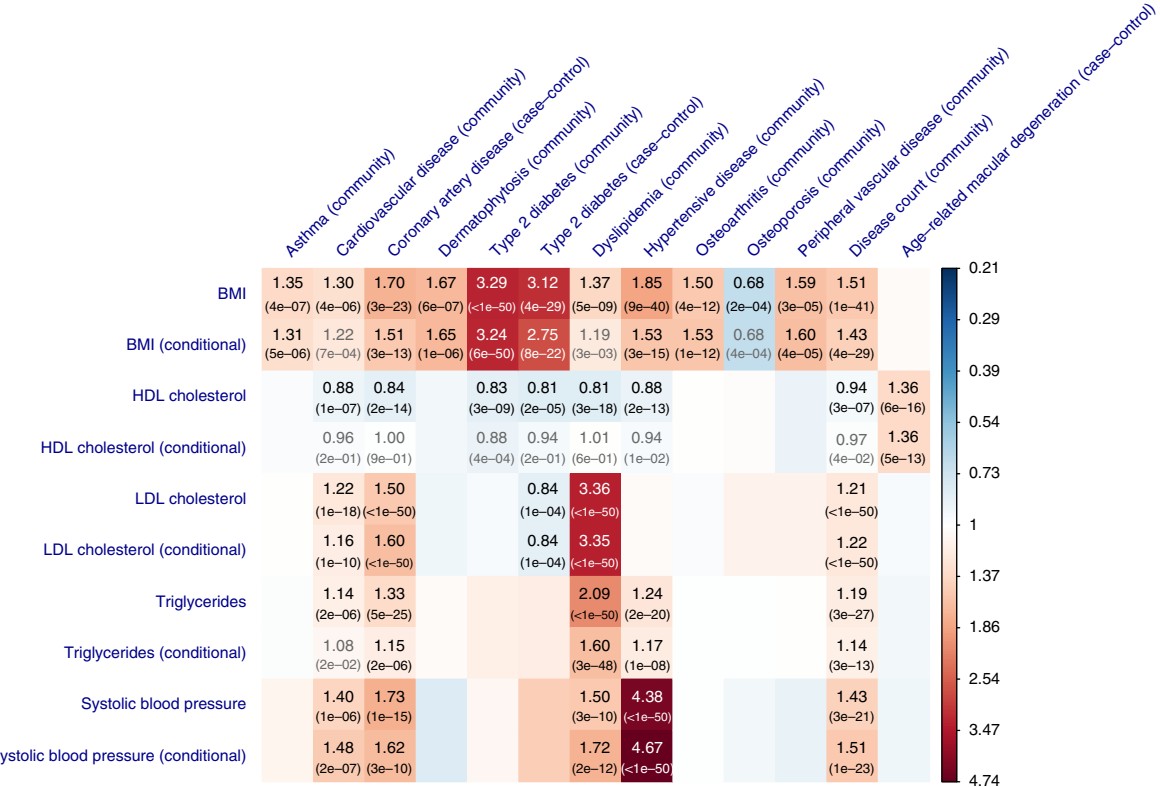

**Fig. 5** GSMR vs. conditional GSMR. Shown are the results from the GSMR analyses compared with those from the conditional GSMR analyses. In the conditional GSMR analysis, the effect size of each risk factor on disease was estimated conditioning on the other risk factors (see Methods for details of the conditional method). "Community": disease GWAS data from a meta-analysis of the two community-based studies. "Case–control": disease GWAS data from independent published case–control studies. In gray are the associations that do not pass the $P$-value threshold $2.2 \times 10^{-4}$ in the conditional analysis

heritability ($h_{\text{SNP}}^2$) for the covariate risk factors, and sampling covariance between SNP effects estimated from potentially overlapping samples, all of which can be computed from summary data (Methods; Supplementary Tables 7–10). Given the similar GSMR results between BMI and WHRadjBMI and between SBP and DBP (Supplementary Fig. 10), we did not include DBP and WHRadjBMI in the conditional analysis to avoid over-correction.

Results from conditional analyses were largely consistent with those from unconditional analyses (Fig. 5; Supplementary Table 11), suggesting that most of the marginal effects are independent of the other risk factors analyzed in this study. Conditioning on the other risk factors, SBP, LDL-c and BMI were the three major risk factors for CAD, BMI was still a large risk factor for T2D and the protective effect of LDL-c on T2D remained largely unchanged (Supplementary Fig. 14). We show above that the GSMR analyses identified significant protective effects of HDL-c against CVD, CAD, T2D and hypertension (Supplementary Fig. 15). However, all the effects became non-significant conditioning on the covariates (i.e., BMI, LDL-c, TG, and SBP), suggesting that the marginal effects of HDL-c on the diseases are not independent of the covariates due to the bidirectional causative associations between HDL-c and the other risk factors as illustrated in Supplementary Fig. 13. It is difficult to distinguish whether the effects of HDL-c on the diseases are mediated or driven by the covariates (Supplementary Fig. 16) because of the complicated association network among risk factors and diseases (Supplementary Fig. 14). Nevertheless, there might be an exception, that is, the association between HDL-c and AMD, because HDL-c is the only risk that showed a significant effect on AMD (OR = 1.36 with $P_{\text{GSMR}} = 5.9 \times 10^{-16}$)

and the effect size remained largely unchanged and highly significant conditioning on the covariates (conditional OR = 1.36 with $P_{\text{GSMR}} = 5.1 \times 10^{-13}$). We conclude that HDL-c is likely to be a direct risk factor for AMD and the effect size is independent of the covariate risk factors analyzed in this study.

Given the estimates from conditional GSMR analyses (Fig. 5; Supplementary Table 11), we could use an approximate approach to calculate the aggregate effect of multiple risk factors on a disease, i.e., $\log(\text{OR}) = \sum [x_i \log(\text{OR}_i)]$. Here is a hypothetical example. If all the risk factors increase by 1 SD (i.e., ~4 kg m$^{-2}$ for BMI, ~1 mmol L$^{-1}$ for LDL-c, ~1 mmol L$^{-1}$ for TG and ~19 mm Hg for SBP), we would have an increased risk of ~2.3-fold to T2D ($e^{1.01-0.17}$), and 4.5-fold to CAD ($e^{0.41+0.47+0.14+0.48}$).

**Effects of other phenotypes on diseases**. Having identified a number of causal associations between seven modifiable risk factors and common diseases, we then sought to test whether there were causative associations between other phenotypes and diseases. We included in the analysis two traits, height[48] and years of schooling[49] (EduYears), for which there were a large number of instruments owing to the large GWAS sample sizes. We selected 811 and 119 near-independent genome-wide significant(GWS) SNPs for height and EduYears, respectively, using the clumping analysis (Methods). The threshold $P_{\text{GSMR}}$ after Bonferroni correction was $7.6 \times 10^{-4}$ correcting for 66 tests. The large number of instruments for height gave us sufficient power to detect a small effect (Fig. 6; Supplementary Table 12; Supplementary Note 5).

Our results also showed that EduYears had protective effects against almost all the diseases (Fig. 6 and Supplementary

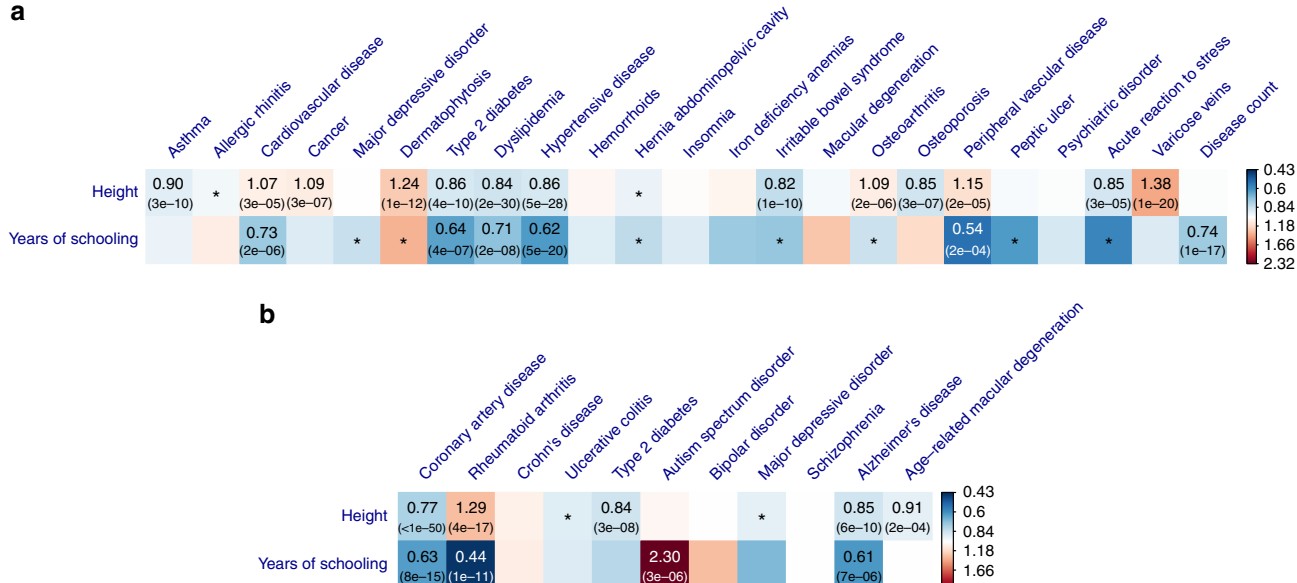

**Fig. 6** Effects of height and educational attainment on common diseases. Shown are the results from GSMR analyses with disease data **a** from a meta-analysis of the GERA and UKB studies and **b** from published independent case–control studies. Colors represent the effect sizes (as measured by odds ratios, ORs) of risk factors on diseases, red for risk effects and blue for protective effects. The significant effects after correcting for multiple testing ($P_{GSMR} < 7.6 \times 10^{-4}$) are labeled with ORs (P-values). The nominally significant effects ($P_{GSMR} < 0.05$) are labeled with "*"

Table 12). It showed protective effect against PVD (OR = 0.54), hypertensive diseases (OR = 0.62), T2D (OR = 0.64), dyslipidemia (OR = 0.71) and CVD (OR = 0.73) in the community data, and RA (OR = 0.44), AD (OR = 0.61) and CAD (OR = 0.63) in the case–control data. It also showed significant protective effect on disease count (OR = 0.74), suggesting that educational attainment is protective for general health outcomes. The protective effect of EduYears against AD is consistent with the observed association from epidemiological studies[50]. On the other hand, however, EduYears showed a strong risk effect on autism spectrum disorder (OR = 2.30) (Supplementary Note 6), which is not influenced by SNP outliers (Supplementary Fig. 17) and consistent with a positive estimate of genetic correlation ($r_g =$ 0.28, SE = 0.038) from a bivariate LD score regression analysis[30].

**Reverse GSMR analysis**. It is important to note that the causative associations identified from the GSMR analyses above are unlikely to be explained by reverse causality for two reasons. First, the individuals used in GWAS for risk factors were independent of the individuals used in GWAS for diseases (the only exception was that the blood pressure GWAS data set was part of the community-based disease GWAS data). Secondly, if the associations presented above are driven by reverse causality, we would expect to see strong association signals of the instruments with the diseases, which is not the case as demonstrated in Supplementary Fig. 18, an idea not too dissimilar to the asymmetry analysis that has been used to infer causality in a previous study[16,22]. Nevertheless, it is interesting to investigate the changes in risk factors after development of the diseases. To do this, we selected instruments for diseases from the disease GWAS data (i.e., GWS SNPs for the disease, hence the instruments used in the reverse-GSMR analysis were distinct from those used in the forward-GSMR analysis). The false positive rate of reverse-GSMR is well calibrated as demonstrated by simulation under the null that there is no reverse effect (Supplementary Fig. 19). We performed a reverse-GSMR analysis of the risk factors and diseases for which there was a significant association in the forward-GSMR analysis above (Supplementary Note 7). We identified

10 significant reverse effects (i.e., the effect of disease on risk factor) in the community data and 4 in the case–control data at a FWER of 0.05 ($P_{reverse-GSMR} < 1.0 \times 10^{-3}$) (Supplementary Table 13). The estimates of reverse effects were very small compared with those of the forward effects. To avoid an underpowered test, we limited the reverse-GSMR analysis to diseases with more than 10 instruments. Given the fact that some of the small estimates of reverse effects were highly significant (Supplementary Table 13), it is unlikely that the large difference in the estimated effect size between the forward and reverse analyses is due to the lack of power in the reverse analysis. We further confirmed by simulation that the GSMR estimate of $b_{xy}$ is unbiased irrespective of the sample size for the exposure (Supplementary Fig. 20). Interestingly, there were two cases where the estimated forward and reverse effects were in opposite directions, i.e., $\hat{b}_{xy(BMI \to T2D)} = 1.19$ and $\hat{b}_{xy(T2D \to BMI)} = -0.07 (P = 3.6 \times 10^{-26})$; $\hat{b}_{xy(BMI \to dyslipidemia)} = 0.32$ and $\hat{b}_{xy(dyslipidemia \to BMI)} = -0.03$ ($P = 2.0 \times 10^{-10}$), meaning that although BMI is risk factor for the two diseases, patients who have developed the diseases may tend to lose weight.

## Discussion

We proposed a flexible and powerful approach that performs a MR analysis with multiple near-independent instruments (i.e., GWS SNPs) to test for causal association between a risk factor (or phenotype) with a disease using summary-level GWAS data from independent studies. We have implemented the method in an R package (URLs). The method and software tool are general and can be applied more broadly to test for causality in other fields such as behavioral sciences. We applied the method to summary data from GWAS of very large sample size, and identified a large number of causal associations between risk factors and common diseases. As the effect sizes of SNPs on risk factor and disease used in the GSMR analysis were from independent GWAS data sets, the effect of risk factor on disease estimated by GSMR was very unlikely to be confounded by environmental factors. The results remain unchanged when we removed SNPs in the major

histocompatibility complex (MHC) region (Supplementary Fig. 21). The result, however, could be biased if there are SNPs that have strong pleiotropic effects on both risk factor and disease. For example, the result for LDL-c and Alzheimer's disease could have been biased due to 16 pleiotropic SNPs (Fig. 4). There are three lines of evidence that our results are not driven by pleiotropy between risk factor and disease. First, as demonstrated in the example above, we have used the HEIDI-outlier approach that removes instruments with strong putative pleiotropic effects (Figs. 3 and 4) and we have confirmed by simulation that the GSMR estimate is unbiased in the presence of LD (Supplementary Table 1; Supplementary Fig. 1). After the HEIDI-outlier filtering, the instruments selected for risk factors did not show strong associations with the diseases except for those highly related diseases and traits (e.g., lipids and dyslipidemia, blood pressures, and hypertensive disease) (Supplementary Fig. 18). Note that the test-statistics decreased slightly after filtering SNPs by HEIDI-outlier (Supplementary Fig. 22), indicating that the result from the analysis with HEIDI-outlier filtering is more conservative. Second, the estimates of $b_{xy}$ were highly consistent with the slopes from Egger regression that are considered to be free of confounding from pleiotropy[13] (MR-Egger) (Supplementary Fig. 23). Note that we used GSMR for the main analyses because in comparison with MR-Egger and inverse-variance weighted method (MR-IVW, equivalent to MR-Egger without intercept)[12], GSMR gains power by taking the sampling variation of $\hat{b}_{zx}$ and $\hat{b}_{zy}$ into account as demonstrated in simulations (Supplementary Fig. 3), and GSMR also has the advantage of accounting for LD among SNPs not removed by the clumping analysis, a property that is important especially when the number of instruments is large. Third, the intercepts from MR-Egger (a significant deviation of the intercept from 0 is evidence for the presence of pleiotropy) were very small relative to the slopes (Supplementary Fig. 24), and there was no inflation in the test-statistics (Supplementary Fig. 24b, c), suggesting that the degree of pleiotropy was negligible if there was any.

We have shown above that our results were not driven by pleiotropy and reverse causality. In some cases, the relationship between a risk factor and a disease could be a mixture of multiple models. For example, we have shown above that BMI had a risk effect on T2D, which has been confirmed by RCT[35], that T2D had a significant reverse effect on BMI and effect size was negative, and that there was a SNP (at the *TCF7L2* gene locus) that appeared to have pleiotropic effects on T2D and BMI (Fig. 3), a mixture model of causality, reverse causality and pleiotropy. In addition, we demonstrated by the conditional GSMR analyses that the mediation effects (i.e., the effect size of a risk factor on disease mediated or driven by other risk factors) are apparently small for most risk factors except for HDL-c (Fig. 5; Supplementary Table 11).

Nevertheless, there are several caveats in interpreting the GSMR results. First, if the exposure is a composite trait that comprises multiple sub-phenotypes, we could not rule out the possibility that the effect of exposure on disease is driven by one of the sub-phenotypes. For instance, we have identified from the GSMR analysis that EduYears had effects on many diseases (Fig. 6). A conservative interpretation is that these are the effects of the genetic component of EduYears (e.g., cognitive ability and personality) on health outcomes. If we express EduYears = g + e, where g is the genetic component of EduYears and e is the residual component that includes environmental influence, then the SNPs identified from GWAS for EduYears are those associated with g rather than e, meaning that the GSMR analysis for Edu-Years was performed on g rather than e and thus did not provide any evidence whether e also has effects on diseases. Therefore, strictly speaking, the causal associations identified in this study

are not definitive and need to be confirmed by follow-up randomized controlled trials (RCTs) in the future, if practical. Second, the effect of a risk factor on disease can be non-linear (e.g., the relationship between BMI and mortality is a U-shaped curve[17], suggesting that both underweight and overweight are risk factors of death) whereas we used a linear approximation to estimate the effect because of the limited information that we had access to from GWAS summary data. Therefore, the $b_{xy}$ estimates need to be interpreted with caution at extremes. Third, although we have identified a large number of associations, we would expect that associations of small effect size would be missed in our study (e.g., the instrument for SBP, SBP, was based on only 28 SNPs). The power can be improved in the future with GWAS results based on larger sample sizes. Fourth, our analyses ignored age-specific and sex-specific effects because of the lack of data from age- and sex-stratified analyses. Lastly, we have shown in a previous study that the SMR test-statistic is slightly deflated due to the use of a Taylor series expansion to compute an approximated sampling variance based on summary data, especially if the association between the instrument and risk factor is not strong enough. We therefore strongly recommend that only SNPs that are associated with the exposure at a genome-wide significance level (i.e., $5 \times 10^{-8}$) should be used in GSMR analysis, and as a rule of thumb advise application only when there are 10 or more independent (e.g., $r^2 < 0.05$) genome-wide significant SNPs.

In summary, we present here summary data-based MR analysis approaches that leverage the large amount of GWAS data from independent studies to detect the effect of a risk factor on disease and assess the effect size conditional on the other risk factors. All the data used in this study were from the public domain, which demonstrates the power of an integrative analysis of existing data to make novel discoveries. The causal associations identified in this study not only provided important candidates to be prioritized in RCTs in the future but also provided fundamental knowledge to understand the biology of the diseases. Our findings of the effects of risk factors on common diseases could have a significant influence on medical research, pharmaceutical industry and public health.

## Methods

**The GSMR method.** Mendelian randomization is a method that uses genetic variants as instrumental variables to test for causative association between an exposure and an outcome[9]. Let z be a genetic variant (e.g., SNP), x be the exposure (e.g., health risk factor) and y be the outcome (e.g., disease). If z is significantly associated with x, the effect of x on y can be estimated using a two-step least squares (2SLS) approach[51]

$$\hat{b}_{xy} = \hat{b}_{zy}/\hat{b}_{zx} \text{ with } \mathrm{var}\left(\hat{b}_{xy}\right) = \mathrm{var}(y)(1 - R^2_{xy})/\left[n\mathrm{var}(x)R^2_{zx}\right],$$

where n is the sample size, $R^2_{xy}$ is the variance in y explained by x, and $R^2_{zx}$ is the variance in x explained z. This analysis requires individual-level data so that the statistical power could be limited if $b_{xy}$ is small. We have previously proposed an approach that only requires summary-level data to estimate $b_{xy}$ so that the power can be greatly improved if $b_{zx}$ and $b_{zy}$ are estimated from independent studies of large sample size[17], i.e., $\hat{b}_{xy} = \hat{b}_{zy}/\hat{b}_{zx}$ with $\mathrm{var}(\hat{b}_{xy}) \approx \frac{b^2_{zy}}{b^2_{zx}}\left[\frac{\mathrm{var}(\hat{b}_{zx})}{b^2_{zx}} + \frac{\mathrm{var}(\hat{b}_{zy})}{b^2_{zy}}\right]$. We called this approach a summary data-based Mendelian randomization (SMR) analysis[17]. We have also shown previously that a SMR analysis using a single genetic variant is unable to distinguish between causality (the effect of SNP on outcome is mediated by exposure) and pleiotropy (the SNP has distinct effects on exposure and outcome). Here, we extend the SMR method to use all the top associated SNPs at a genome-wide significance level for the exposure as instrumental variables to test for causality. We call this method a generalized SMR (GSMR) analysis. The basic idea of GSMR is that if x is causal for y, any SNP associated with x will have an effect on y, and the expected value of $\hat{b}_{xy(i)}$ at any SNP i will be identical in the absence of pleiotropy. Let m be the number of GWS top SNPs associated with x after clumping. We have $\hat{\mathbf{b}}_{xy} = \left\{\hat{b}_{xy(1)}, \hat{b}_{xy(2)}, \cdots, \hat{b}_{xy(m)}\right\}$ with $\hat{b}_{xy(i)} = \hat{b}_{zy(i)}/\hat{b}_{zx(i)}$, and $\hat{\mathbf{b}}_{xy} \sim N(\mathbf{1}\mathbf{b}_{xy}, \mathbf{V})$ where $\mathbf{1}$ is an $m \times 1$ vector of ones and $\mathbf{V}$ is the variance-covariance matrix of $\hat{\mathbf{b}}_{xy}$. We have derived previously that the ij-th element of $\mathbf{V}$ is

$$\text{cov}\left(\hat{b}_{xy(i)}, \hat{b}_{xy(j)}\right) \approx \frac{r}{b_{zx(i)} b_{zx(j)}} \sqrt{\text{var}\left(\hat{b}_{zy(i)}\right) \text{var}\left(\hat{b}_{zy(j)}\right)} + b_{xy(i)} b_{xy(j)}$$

$$\left[\frac{r \sqrt{\text{var}\left(\hat{b}_{zx(i)}\right) \text{var}\left(\hat{b}_{zx(j)}\right)}}{b_{zx(i)} b_{zx(j)}} - \frac{\text{var}\left(\hat{b}_{zx(i)}\right) \text{var}\left(\hat{b}_{zx(j)}\right)}{b_{zx(i)}^2 b_{zx(j)}^2}\right]$$

, where subscripts $i$ and $j$ represent SNP $i$ and $j$, respectively, $r$ is LD correlation between the two SNPs (not available in the summary data but can be estimated from a reference sample with individual-level genotypes). The $i$-th diagonal element of $\mathbf{V}$ is $\text{var}\left(\hat{b}_{xy(i)}\right) = b_{xy(i)}^2 \left[\frac{\text{var}\left(\hat{b}_{zx(i)}\right)}{b_{zx(i)}^2} + \frac{\text{var}\left(\hat{b}_{zy(i)}\right)}{b_{zy(i)}^2} - \frac{\text{var}^2\left(\hat{b}_{zx(i)}\right)}{b_{zx(i)}^4}\right]$. Therefore, we can estimate $b_{xy}$ from all the instruments using the generalized least squares approach as $\hat{b}_{xy} = \left(\mathbf{1}'\mathbf{V}^{-1}\mathbf{1}\right)^{-1} \mathbf{1}'\mathbf{V}^{-1}\hat{\mathbf{b}}_{xy}$ with $\text{var}\left(\hat{b}_{xy}\right) = \left(\mathbf{1}'\mathbf{V}^{-1}\mathbf{1}\right)^{-1}$. The statistical significance of $\hat{b}_{xy}$ can be tested by $T_{GSMR} = \hat{b}_{xy}^2 / \text{var}\left(\hat{b}_{xy}\right)$ which follows a $\chi^2$ distribution with 1 degree of freedom. Note that because logOR is free of ascertainment bias (i.e., the bias due to a higher proportion of cases in the sample than in the general population), the method can be applied to disease data from case–control studies, and the estimate of $b_{xy}$ should be interpreted as that of the general population.

**Removal of pleiotropic SNPs by HEIDI-outlier**. We have shown above that under a causal model the expected value of $\hat{b}_{xy}$ estimated at any of the SNP instruments is identical in the absence of pleiotropy. If there are SNPs that have pleiotropic effects on $x$ and $y$, $\hat{b}_{xy}$ estimated at these SNPs will deviate from the expected value under a causal model, and hence will present as outliers. There have been methods to assess the sensitivity of an MR analysis to detect pleiotropy[52]. These methods, however, do not account for possible LD between SNPs nor the sampling errors in the estimated effect sizes of the instruments on the exposures. We previously proposed an approach (heterogeneity in dependent instrument, HEIDI) to test for heterogeneity in $b_{xy}$ estimated at multiple correlated instruments[17]. Here, we extend this approach to detect heterogeneity in $b_{xy}$ estimated at $m$ near-independent instruments (note that the method accounts for remaining LD not removed by clumping). The basic idea is to test where there is a significant difference between $b_{xy}$ estimated at an instrument $i$ (i.e., $b_{xy(i)}$) and $b_{xy}$ estimated at a target SNP that shows a strong association with the exposure. The power of detecting heterogeneity increases with the strength of association between the target SNP and exposure. However, we cannot simply choose the top exposure-associated SNP because sometimes when a SNP has an extremely strong effect on the exposure, it is also likely to be a pleiotropic outlier (e.g., the top LDL-associated SNP at the *APOE* locus shows a very strong pleiotropic effect on Alzheimer's disease, as shown in Fig. 4). Therefore, to increase the robustness of the HEIDI-outlier test, we examine the distribution of $\hat{b}_{xy}$ as a function of $-\log 10(P\text{-value})$ for $\hat{b}_{zx}$ and choose the SNP that shows the strongest association with the exposure in the third quintile of the distribution of $\hat{b}_{xy}$ to avoid choosing an extreme pleiotropic outlier as the target SNP. If we define $d_i = b_{xy(i)} - b_{xy(top)}$, we will have $\text{var}\left(\hat{d}_i\right) \text{var}\left(\hat{b}_{xy} - \hat{b}_{xy(top)}\right) = \text{var}\left(\hat{b}_{xy(i)}\right) + \text{var}\left(\hat{b}_{xy(top)}\right) - 2\text{cov}(\hat{b}_{xy(i)}, \hat{b}_{xy(top)})$, where $\text{cov}\left(\hat{b}_{xy(i)}, \hat{b}_{xy(top)}\right) = \frac{r_i \sqrt{\text{var}(\hat{b}_{zy(i)}) \text{var}(\hat{b}_{zy(top)})}}{b_{zx(i)} b_{zx(top)}} + b_{xy(i)} b_{xy(top)} \left[\frac{r_i \sqrt{\text{var}(\hat{b}_{zx(i)}) \text{var}(\hat{b}_{zx(top)})}}{b_{zx(i)} b_{zx(top)}} - \frac{\text{var}(\hat{b}_{zx(i)}) \text{var}(\hat{b}_{zx(top)})}{b_{zx(i)}^2 b_{zx(top)}^2}\right]$, and $r$ is the LD correlation between the two SNPs (estimated from a reference sample with individual-level genotypes). We can test the deviation of each SNP from the causal model using the $\chi^2$-statistic $T = \hat{d}_i^2 / \text{var}(\hat{d}_i)$, and remove the SNPs with $P$-values < 0.01. We call this approach HEIDI-outlier. We choose a relatively less stringent $P$-value threshold for the HEIDI-outlier analysis because even if a causal signal is detected as pleiotropy and eliminated from the analysis, it will only affect the power rather than the false positive rate or biasedness of the GSMR analysis. To retain as much power as possible to detect heterogeneity, we use a modest threshold 0.01. This means that even if there is no pleiotropic outlier, we will remove only ~1% of the instruments by chance, which is very unlikely to result in a substantial decrease in power of the subsequent GSMR analysis.

**Multi-trait conditional GWAS analysis using summary data**. To test whether the effect of a risk factor ($x_0$) on a disease ($y$) depends on other risk factors ($\mathbf{x} = \{x_1, x_2, ..., x_i\}$), we usually perform a joint analysis based on the model below

$$y = x_0 b_0 + \mathbf{x}\mathbf{b}_{xy} + e,$$

where $b_0$ is the effect of $x_0$ on $y$, $\mathbf{b}_{xy} = \{b_{x,y}\}$ is a $t$-length vector with $b_{x,y}$ being the effect of a covariate $x_i$ on $y$, and $e$ is the residual. Such an analysis is equivalent to a two-step analysis with the first step to adjust both $x_0$ and $y$ by $\mathbf{x}$ and the second step to estimate the effect of adjusted $x_0$ on adjusted $y$. We therefore can estimate the effect size of $x_0$ on $y$ accounting for $\mathbf{x}$ by a GSMR analysis using SNP effects on $x_0$ and $y$ conditioning on $\mathbf{x}$.

The conditional GWAS analysis usually requires individual-level genotype and phenotype data, which are not always available. Here, we propose a method to perform an approximate multi-trait-based conditional GWAS analysis that only requires summary data. Since GWAS summary data for risk factors and disease are often from multiple independent studies, the analysis has to be performed

conditioning on the genetic values of the covariate risk factors (denoted by $\mathbf{g}_x = \{g_{x_1}, g_{x_2}, ..., g_{x_t}\}$), where the genetic value is defined as the aggregated effect of all SNPs on a phenotype accounting for LD. Following the method that uses GWAS summary data to perform a multi-SNP-based conditional and joint analysis (GCTA-COJO)[53], the SNP effect on the disease accounting for $\mathbf{g}_x$ can be expressed as

$$\hat{b}_{zy}|\hat{\mathbf{b}}_{xy} = \hat{b}_{zy} - \hat{\mathbf{b}}_{zx}^t \hat{\mathbf{b}}_{xy},$$

where $\hat{b}_{zy}$ is the SNP effect on the disease on the logit scale (i.e., logOR), $\hat{\mathbf{b}}_{xy}$ is a $t$-length vector with the $i$-th element $\hat{b}_{x,y}$ being the effect of $g_{x_i}$ on the disease when all the covariates are fitted jointly, and $\hat{\mathbf{b}}_{zx}$ is a $t$-length vector of SNP effects on $\mathbf{x}$. For the ease of derivation, we assume each covariate has been standardized with mean 0 and variance 1 (note that the method can be applied to data on the original scale without standardization). We know from previous studies[53] that the joint effects of $\mathbf{g}_x$ on y ($\mathbf{b}_{xy}$) can be transformed from the marginal effects ($\beta_{xy}$), i.e.,

$$\mathbf{b}_{xy} = \mathbf{D}^{-\frac{1}{2}}\mathbf{R}_x^{-1}\mathbf{D}^{\frac{1}{2}}\boldsymbol{\beta}_{xy},$$

where $\mathbf{R}_x = \{r_{g(x_i, x_j)}\}$ is a $t \times t$ matrix with $r_{g(x_i, x_j)}$ being the genetic correlation between covariates $i$ and $j$, $\mathbf{D}$ is a $t \times t$ diagonal matrix with the $i$-th diagonal element $h_{\text{SNP}(x_i)}^2$ being the SNP-based heritability for the $i$-th covariate. We can estimate $h_{\text{SNP}(x_i)}^2$ and $r_{g(x_i, x_j)}$ from GWAS summary data using the LDSC approaches[30,54], and estimate $\beta_{x_i y}$ by GSMR.

The sampling variance of $\hat{b}_{zy}|\hat{\mathbf{b}}_{xy}$ is approximately

$$\text{var}\left(\hat{b}_{zy}|\hat{\mathbf{b}}_{xy}\right) = \text{var}\left(\hat{b}_{zy}\right) + \hat{\mathbf{b}}_{xy}^t \mathbf{V}_{zx} \hat{\mathbf{b}}_{xy} - 2\hat{\mathbf{b}}_{xy}^t \text{cov}\left(\hat{b}_{zy}, \hat{\mathbf{b}}_{zx}\right),$$

where $\mathbf{V}_{zx} = \text{var}(\hat{\mathbf{b}}_{zx})$, and $\text{cov}\left(\hat{b}_{zy}, \hat{\mathbf{b}}_{zx}\right)$ is a $t$-length vector with the $i$-th element $\text{cov}\left(\hat{b}_{zy}, \hat{b}_{zx_i}\right)$ being the covariance between $\hat{b}_{zy}$ and $\hat{b}_{zx_i}$. We know from our previous study[17] that $\text{cov}\left(\hat{b}_{zy}, \hat{b}_{zx_i}\right) = \rho_{x_i y} r_{p(x_i, y)} \sqrt{\text{var}\left(\hat{b}_{zx_i}\right) \text{var}\left(\hat{b}_{zy}\right)}$ where $\rho_{x_i y}$ is the proportion of sample overlap between $x_i$ and $y$ and $r_{p(x_i, y)}$ is the phenotypic correlation between $x_i$ and $y$. In special cases, if $y$ and $\mathbf{x}$ are observed in the same sample, $\text{var}\left(\hat{b}_{zy}|\hat{\mathbf{b}}_{xy}\right) = \text{var}\left(\hat{b}_{zy}\right) - \hat{\mathbf{b}}_{xy}^t \mathbf{V}_{zx} \hat{\mathbf{b}}_{xy}$, and if there is no sample overlap between $y$ and $\mathbf{x}$, $\text{var}\left(\hat{b}_{zy}|\hat{\mathbf{b}}_{xy}\right) = \text{var}\left(\hat{b}_{zy}\right) + \hat{\mathbf{b}}_{xy}^t \mathbf{V}_{zx} \hat{\mathbf{b}}_{xy}$. More generally, if there is a sample overlap between $y$ and $\mathbf{x}$, $\rho_{x_i y} r_{p(x_i, y)}$ can be approximated by the intercept of a bivariate LDSC analysis between $x_i$ and $y$ (ref. [30]). $\mathbf{V}_{zx}$ is the sampling variance-covariance of $\hat{\mathbf{b}}_{zx}$ with the $ij$-th element $\text{cov}\left(\hat{b}_{zx(i)}, \hat{b}_{zx(j)}\right) = \rho_{x_i x_j} r_{p(x_i, x_j)} \sqrt{\text{var}\left(\hat{b}_{zx_i}\right) \text{var}\left(\hat{b}_{zx_j}\right)}$, where $\rho_{x_i x_j} r_{p(x_i, x_j)}$ can also be approximated by the intercept of a bivariate LDSC analysis between $x_i$ and $x_j$. The multi-trait-based conditional GWAS test can be performed using the test-statistic $T_{\text{cond}} = \left(\hat{b}_{zy}|\hat{\mathbf{b}}_{xy}\right)^2 / \text{var}\left(\hat{b}_{zy}|\hat{\mathbf{b}}_{xy}\right)$. We call this approach mtCOJO (multi-trait-based conditional and joint analysis), and have demonstrated the accuracy of the approximation by simulation (Supplementary Fig. 5). Note that since the estimate of $\beta_{x_i y}$ is free of confounding from shared environmental or genetic effects that are not correlated with the valid instruments, our estimate of conditional effect does not suffer from the bias described in Aschard et al.[23], as confirmed by simulation (Supplementary Fig. 6). We have implemented mtCOJO in the GCTA software package (URLs).

**GWAS data for risk factors and diseases**. We used nine risk factors as exposures for the GSMR analysis. These include seven health risk factors, i.e., body mass index (BMI), waist-to-hip ratio adjusted by BMI (WHRadjBMI), HDL cholesterol (HDL-c), LDL-cholesterol (LDL-c), triglyceride (TG), systolic blood pressure (SBP) and diastolic blood pressure (DBP), and two additional phenotypes (height and educational attainment) that had a large number of instruments. We conducted GWAS analyses for SBP and DBP using data from the UK Biobank[27] (UKB) (see below for details of the UKB data). GWAS summary data for the other traits were from published studies (Supplementary Table 3). We re-calculated $\hat{b}_{zx}$ from z-statistics ($z_{zx}$) using the method described in Zhu et al.[17] so that $\hat{b}_{zx}$ could be interpreted in SD units (i.e., $\hat{b}_{zx} = \frac{z_{zx}}{\sqrt{2p(1-p)(n+z_{zx}^2)}}$ with $p$ being the allele frequency and $n$ being the sample size). We then applied the clumping algorithm in PLINK[28] to select near-independent GWS SNPs for each trait ($r^2$ threshold = 0.05, window size = 1 Mb and $P$-value threshold = $5 \times 10^{-8}$) using the 1000G-imputed ARIC data[33] ($n = 7,703$ unrelated individuals) as the reference for LD estimation. As the statistical power of the GSMR analysis increases as the number of instruments, we performed the clumping analysis repeatedly for the SNPs in common between each pair of risk factor and disease data sets to maximize the number of instruments.

GWAS data for 22 common diseases were from two community-based studies, i.e., Genetic Epidemiology Research on Adult Health and Aging[29] (GERA) and UKB pilot phase[27]. There were 60,586 individuals of European ancestry in the GERA data. We cleaned the GERA genotype data using the standard quality control (QC) filters (excluding SNPs with missing rate ≥0.02, Hardy–Weinberg equilibrium test $P$-value ≤ $1 \times 10^{-6}$ or minor allele count < 1, and removing

individuals with missing rate ≥0.02), and imputed the genotype data to the 1000G using IMPUTE2[55]. We used GCTA[56] to estimate the genetic relationship matrix (GRM) of the individuals using a subset of the imputed SNPs (minor allele frequency, MAF ≥0.01 and imputation INFO score ≥0.3 and in common with those in the HapMap phase 3, HM3), and computed the first 20 principal components (PCs) from the GRM. We removed one of each pair of individuals with estimated genetic relatedness ≥0.05 and retained 53,991 unrelated individuals for analysis. Individual-level ICD-9 codes were not available in dbGaP but had been classified into 22 common diseases (Supplementary Table 4). The disease status was coded as 0 (unaffected) and 1 (affected). We added an additional trait "disease count" (a count of the number of diseases affecting each individual) as a crude measure of general health status of each individual. We then performed a genome-wide association analysis for each of the 23 phenotypes with age, gender, and the first 20 PCs fitted as covariates. The MHC region is often removed from the analysis in previous studies, mainly because of the complicated LD structure in this region. In this study, we did not remove this region because we use a set of near-independent SNPs as instruments after LD clumping.

Genotype data from UKB pilot phase had been cleaned and imputed to a combined reference panel of 1000G and UK10K (see UKB documentation for details about QC and imputation). We included in the analysis only the individuals of European ancestry. Similarly as above, we computed the GRM and the first 20 PCs based on the HM3 SNPs with MAF ≥0.01 and imputation INFO score ≥0.3, and retained 108,039 unrelated individuals (GRM threshold of 0.05) for analysis. Individual-level ICD-10 codes were available in the UKB data. To match the diseases in GERA, we classified the phenotypes into 22 common diseases by projecting the ICD-10 codes to the classifications of ICD-9 codes in GERA taking into account self-reported disease status (Supplementary Table 4). We also added the trait "disease count". We then conducted genome-wide association analyses for the 23 phenotypes using the same approach as above.

**URLs**. GSMR R package: http://cnsgenomics.com/software/gsmr/
mtCOJO: http://cnsgenomics.com/software/gcta/#mtCOJO
SMR: http://cnsgenomics.com/software/smr
PLINK: http://pngu.mgh.harvard.edu/~purcell/plink/
PLINK2: https://www.cog-genomics.org/plink2
GCTA: http://cnsgenomics.com/software/gcta
LDSC: https://github.com/bulik/ldsc

**Data availability**. The summary-level GWAS data from the meta-analyses of GERA and UKB are available at http://cnsgenomics.com/data.html. All the other data sets used in this study are from the public domain. The software tools are available at the URLs above.

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

## Acknowledgements

This research was supported by the Australian Research Council (DP160101343), the Australian National Health and Medical Research Council (1107258, 1078901, 1078037, 1056929, 1048853, and 1113400), the Sylvia & Charles Viertel Charitable Foundation (Senior Medical Research Fellowship), and the Danish National Research Foundation (Niels Bohr Professorship). This study makes use of data from dbGaP (Accession Numbers: phs000090.v3.p1 and phs000674.v2.p2) and UK Biobank Resource (Application Number: 12514). A full list of acknowledgements to these data sets can be found in Supplementary Note 8.

## Author contributions

J.Y. conceived and designed the study. J.Y. and Z.Z. derived the theories. Z.Z. performed simulations and statistical analyses under the assistance and guidance from Zhili.Z., F.Z., Y.W., M.T., R.M., M.R.R., J.J.M., P.M.V., N.R.W. and J.Y. Z.Z. and Zhili.Z. developed the software tool. J.Y. and Z.Z. wrote the manuscript with the participation of all authors. All authors reviewed and approved the final manuscript.

## Additional information

**Competing interests:** The authors declare no competing financial interests.

