## [Peer Review File · Nature Communications]

Reviewers' comments:

Reviewer #1 (Remarks to the Author):

This manuscript applies Mendelian Randomization (MR) to understand the impact of several risk factors on complex traits. The authors employ a few variations on MR to get better estimates of the causal effects - 1) detection and removal of outlier SNPs that may be indicative of pleiotropy 2) integration of multiple independent SNPs into a better estimate of the MR statistic and 3) conditional analysis to control for other covariates. I found it to be an interesting paper to read, relevant to current approaches in MR and GWAS, but not exceptionally novel given related work - rather a collection of variations that hopefully improve the results, but need some greater demonstration.

One major omission in this work is a demonstration of the improvement by each of these variations, or a comparison to other related approaches (such as Pickrell et al). The reader needs to be shown that these are better methods for getting more accurate estimates of causal relationships, and to use for future studies. This ideally would include comparison to results from randomized trials (if quantitative results are available), and simulation analysis. This is important to understanding the contribution.

A particular place where more analysis would help is in the analysis of HEIDI-outlier. They acknowledge it is not a "fail-safe" but the reader is left with little sense of how many pleiotropic effects (or what types/effect sizes) would escape detection. The mention a false positive rate of 0.01 given their lax p-value threshold, but more importantly, what is the false negative rate?

Even though the reverse causality tests generally revealed smaller effect sizes, they did find a reasonable number of hits even a strict FWER - this merits more explanation. Do these indicate false positives (more than the p-values would indicate, and thus potentially aren't calibrated), or are they suggesting these are actually true reverse effects (in addition to the forward effects?)?

Other comments:

Can this be explained a little further (from Online Methods): "where there is a significant difference between b_{xy} estimated at an instrument i (i.e. $b_{xy}(i)$) and b_{xy} estimated at the SNP that shows the strongest association with exposure in the third quintile of the $b^{\wedge}xy$ distribution" -- why the third quintile?

The wording of this sentence is a bit convoluted: "We have previously shown that the power of MR could be greatly improved by a flexible analysis of summary-level GWAS data for exposure (e.g. risk factor) and outcome (e.g. disease) from two samples of large sample size (summary-data-based MR, SMR), and applied the SMR method to test if the effects of genetic variants on a phenotype are mediated by gene expression"

"estimate the effect of a risk factor on disease conditioning on the genetic values of other risk factors" - For a general audience should define "genetic value" the first time it's used.

In the following sentences, they seem to contradict themselves. First saying there is likely mediation, then saying they can't really distinguish. I would tone down the first sentence: "suggesting that the marginal effects of HDL-c on diseases were likely to be mediated or driven by the covariate risk factors because of the complex bidirectional causative associations between HDL-c and the other risk factors as illustrated in Supplementary Fig. 9. It is difficult to distinguish the direction of mediation, i.e. whether HDL-c is a mediating (model I: other risk factors -> HDL-c -> disease) or driving (model II: HDL-c -> other risk factors -> disease) factor (Supplementary Fig. 13)."

I would normally say that MR *uses* instrumental variables, not that MR *is* an instrumental variable. (from Introduction)

Reviewer #2 (Remarks to the Author):

The authors proposed three summary-data-based methods in this manuscript: 1) to estimate and test the mediation effect of risk factor on disease status ($b_{\{xy\}}$) using multiple genetic variants as instruments, an extension of their previous SMR method 2) to detect outliers in the estimates of mediation effect which are likely caused by pleiotropy, i.e. not by mediation / causality 3) to estimate and test the mediation effect conditional on other risk factors. They applied their methods to multiple GWAS summary association data set to test for causal relationships between pairs of traits and diseases. Overall, I find the methods robust and useful addition to the community. However, more simulations and more rigorous analyses of real data are needed to demonstrate the robustness of their methods and validity of their conclusions.

Major comments:

1) The authors developed GSMR to meta-analyze the estimates of effect of risk factor on disease status from multiple SNPs ($b_{\{zx,i\}}$) in a fixed-effect framework. This method models correlation between SNPs (LD) in the variance, and therefore is able to provide an unbiased and efficient estimate of the mediation effect. However, although the authors claim that their method can account for remaining LD after clumping, all of their simulations are based on independent SNPs – SNPs were drawn independently from the binomial. It would greatly strengthen the manuscript if the author could perform simulations where the SNPs are not entirely independent from each other and then show that their method is still unbiased in the presence of LD.

2) In the null simulation to test the unbiasedness of GSMR, the authors did not include any pleiotropic / direct effect of genetic effect on the disease status. For example, it could be the case that $b_{\{xy\}}$ is zero, but multiple $b_{\{zy\}}$ is not zero for both the exposure and the disease status at some of the SNPs. This could, in principle, results in bias estimate of $b_{\{xy\}}$, in the scenario where there is LD among the SNPs – the estimate of $b_{\{xy,i\}}$ at each SNPs are likely non-zero and highly correlated. It would greatly strengthen the manuscript if the authors could perform simulations where there is indeed pleiotropic / direct effect of genetic variants on the disease status. Also, although the authors demonstrated that the power of GSMR is not significantly affected by HEIDI-outlier filtering, an assessment of HEIDI-outlier in detecting the SNPs with pleiotropic effect would be very helpful.

3) The authors showed that their method is more powerful than Egger regression. However, this is not an entirely fair comparison, since Egger regression accounts for pleiotropy by incorporating an intercept term in the second regression step, effectively estimating two parameters instead of one as the case for GSMR. It's curious whether GSMR is still more powerful than Egger regression when there is indeed pleiotropy. Again, all the simulations performed in this study (as described in supplementary note) do not seem to make the assumption that there is indeed pleiotropy. An explanation of why GSMR is more powerful than Egger regression would also be helpful.

4) In the absence of LD, the inverse-variance weighted approach to estimate the mediation effect should be equivalent to the GSMR approach (as shown by the equations at the top of page 23) and Egger regression. It's curious to me why GSMR has more power than IVW. An explanation would be helpful.

5) The authors analyzed many diseases that are case-control traits, which are often studied on ascertained samples, i.e. more cases are collected than in the general population to increase the chance of detecting the causal variants. The exposures, which are quantitative traits, on the other hand, are more likely studied on randomly collected (i.e. not ascertained) samples. The mismatch

between the two types of samples will likely induce an overestimate of the mediation effect. The authors should elaborate more on how ascertainment could bias their causal inference.

6) It's not clear whether the authors removed genetic variants in the HLA region before their analyses. If the author included SNPs in the HLA, an explanation would be helpful.

Minor comments:

1) The authors estimate $b_{\{xy\}}$ by meta-analyzing $b_{\{xy,i\}}$ of each SNP under a fixed-effect framework. Another approach is to assume the true $b_{\{xy\}}$ has a distribution, i.e. a random-effect framework. It's curious whether using a random-effect framework would be more appropriate as each SNP in the GWAS is likely tested under different sample sizes.

2) The authors showed that reverse causation likely exist for BMI and T2D although the effect of T2D on BMI is much smaller. It's worth noting here that the sample sizes of GWASs on BMI and T2D are very different. Therefore, there can be biases in the number of GWS selected (e.g. number of GWAS for BMI likely larger than the number of GWS for T2D) in the estimation $b_{\{xy\}}$, which can in turn results in biases in $b_{\{xy\}}$ in both the forward and reverse direction. The authors should discuss this as potential bias of their analyses.

3) It would be helpful to provide an estimate of the remaining LD (e.g. the LD score) after applying the clumping step in real data analyses.

4) I would run the causal inference methods described in citation 16 to confirm / support the findings discovered in this manuscript.

5) Adjusting for heritable covariates could induce bias in GWAS (see Aschard et al. AJHG 2015). Since most of the GWAS summary association data are likely adjusted for covariates, this could induce biases in estimating $b_{\{xy\}}$. Similarly, applying conditional analyses using summary data could result in false causative association as well.

Reviewer #3 (Remarks to the Author):

Zhu et al. propose the method GSMR to estimate "causal" (with assumptions) relationships between traits from GWAS summary statistics. They extend their previous SMR statistic to estimate and test a Mendelian Randomization across multiple correlated instruments. They propose an outlier test to identify and remove SNPs that deviate from the overall MR trend, as well as a novel conditional analysis. Because the method only requires summary-level data, it was applied to a large number of risk factors and disease traits. They observe a causal effect of BMI on many traits (including, interestingly, overall disease count), as well as intriguing causal inferences between height, educational attainment and other traits.

Overall, the work aims to address a specific and important problem, proposes multiple methods, and describes interesting results. The manuscript is concise and easy to read. The conditional GSMR idea, in particular, is novel and informative. However, given that MR is now a mature research area with many competing methods, it is not clear what advance these methods offer over the most cutting-edge work, which is not cited or discussed very extensively. Moreover, there are some issues with the SMR statistic and the simulation framework that left me unconvinced about the validity of the method. The results are interesting and of potential value to the field, but there are too many methodological gaps in the current form.

Major Comments

* Recent MR methodology should be cited and compared to more thoroughly. Several methods to

deal with correlated instrumental variables exist, of which weighted generalized linear regression (Burgess et al. Stat Med 2016 [PMID 26661904]) appears to be very similar in spirit to GSMR. What is the advance of this approach over the methods described in Burgess et al. and what is the relative performance of the methods? The approach of Pickrell et al. Nat Genet 2016 - perhaps the most recent high-profile MR paper - is cited but GSMR needs to be put in context to the causal inference in that paper (which analyzes many of the same traits) or compared by simulation. The same comments apply to the HEIDI outlier test: multiple summary-based sensitivity approaches are discussed in, for example, (Burgess et al. Epi 2017 [PMID 27749700]). How does HEIDI-outlier compare to those approaches?

* The derivation for $\text{var}(b_{xy})$ is approximate (pg.22) and is not well calibrated, casting doubt on the calibration of the subsequent statistics. Consider the following R snippet as a quick example:

```
set.seed(0)
chisqzy = rchisq(10e4,df=1,ncp=0)
chisqzx = rchisq(10e4,df=1,ncp=0)
smrstat = chisqzy * chisqzx / (chisqzy + chisqzx)
cat( mean(pchisq(smrstat,1) < 0.05) , '\n' )
```

In this null simulation 10% of statistics come up as significant at $P < 0.05$. Only after substantially increasing the non-centrality parameter for one of the traits does the empirical α start to approach 5%. This is a serious problem that the reader should be made aware of and I did not find any discussion of it in the manuscript. Especially since the actual causal estimate is identical to existing methods (Fig S16) so all of the power is coming from this variance estimator. This is the fundamental statistic used in all other tools so poor calibration has implications for all of the results. Please include a discussion of this issue and recommendation for how to avoid bias, and assure the reader that mis-calibration is not the reason GSMR performs better than existing methods.

* All of the simulations in Supp Note 1 use independent SNPs drawn from a binomial distribution so the impact of LD is never actually evaluated. An LD correlation matrix is described in the Supplement ("In addition, we simulated 5,000 individuals in sample #3 (n_3) to calculate LD correlation matrix.") but should only be contributing noise if the SNPs come from independent distributions. Please include thorough simulations with realistic population LD and reference panels (as in Supp Note 3 for example).

* The GSMR method is presented as a generalization of MR that accounts for LD, and so I had expected it to be applied to most/all SNPs in the data. However, only genome-wide significant SNPs with highly strict LD pruning were used. It's not at all clear why these restrictions are necessary and they severely undercut the novelty of the approach since the remaining SNPs are nearly free of LD. Why throw away so much data? Given that this is such a key methodological point, I urge the authors to clearly explain and justify how SNPs should be selected for inclusion in the analysis to maximize power (this could be addressed using realistic LD simulations suggested in the previous comment).

* There's also a broader question of what advantages this approach has over cross-trait LD-score regression (which was run on all pairs of traits anyway). The paper of Bulik-Sullivan et al. 2015 Nat Genet showed that cross-trait LDSC is asymptotically equivalent to the 2SLS Mendelian randomization estimate from the same set of variants, and confounding from pleiotropy affects both methods. So is there a clear advantage to using the GSMR statistics? Are there any instances where GSMR results are significantly different from cross-trait LDSC results? Is GSMR expected to have better power for certain disease architectures? I can see how the bi-directional GSMR approach gives you additional insights into causality, but is this better than running a sort of "bi-directional" LDSC on the top X% of SNPs from each trait?

Minor Comments

* For HEIDI-outlier please explain why the third quintile is used to define $b_{xy}(\text{top})$ and how the choice of quintile impacts the power and calibration of the method. Can the authors guarantee that using HEIDI-outlier to remove poor fitting SNPs and then running GSMR is always over-conservative?

* The multi-trait conditioning requires h^2 and r^2 over the targeted SNPs, but uses LDSC estimates which are from all common variants. Is there a misspecification if the distribution of effect sizes is something like spike + slab and top hits have different h^2 and r^2 from the rest of the variants?

* The reverse GSMR statistics being less significant than forward GSMR is used as evidence of little pleiotropy: "Second, if the results were driven by pleiotropy, we would expect the estimates of b_{xy} from reverse GSMR comparable with those from GSMR, which is not what we observed". However, is this not strongly effected by differences in power between the two studies determining which instruments get selected? The GIANT and Edu traits are some of the largest GWAS in existence so it's not entirely surprising that associations in the reverse direction are weaker and may not be sufficient to rule out pleiotropy or reverse-causality.

Reviewers' comments:

We thank the constructive comments from the three reviewers, which have significantly improved our manuscript. We have responded to all the reviewers' comments point-by-point below in this document (in blue) and have highlighted all the relevant changes in yellow in the revised manuscript.

During the revision process, we updated our multi-trait-based conditional method to account for potential sample overlaps among data sets. We have re-run the conditional analyses using the updated method. The results remain mainly unchanged.

Reviewer #1 (Remarks to the Author):

This manuscript applies Mendelian Randomization (MR) to understand the impact of several risk factors on complex traits. The authors employ a few variations on MR to get better estimates of the causal effects - 1) detection and removal of outlier SNPs that may be indicative of pleiotropy 2) integration of multiple independent SNPs into a better estimate of the MR statistic and 3) conditional analysis to control for other covariates. I found it to be an interesting paper to read, relevant to current approaches in MR and GWAS, but not exceptionally novel given related work - rather a collection of variations that hopefully improve the results, but need some greater demonstration.

One major omission in this work is a demonstration of the improvement by each of these variations, or a comparison to other related approaches (such as Pickrell et al). The reader needs to be shown that these are better methods for getting more accurate estimates of causal relationships, and to use for future studies. This ideally would include comparison to results from randomized trials (if quantitative results are available), and simulation analysis. This is important to understanding the contribution.

Re: We thank the reviewer for the suggestion. We have compared GSMR with the prevailing methods that use GWAS summary data to infer causality, e.g. the inverse-variance weighted (IVW) Mendelian Randomisation (MR) method, linear regression based MR method (i.e. MR-Egger), and the Pickrell maximum likelihood method (Pickrell 2016 Nat Genet).

Previous studies have shown by theory and simulation (Burgess et al. 2013 Genet Epidemiol; Bowden et al. 2015 Int J Epidemiol) that MR-IVW is equivalent to MR-Egger with intercept 0.

The Pickrell methods are not based on the MR framework. One method is to use the correlation in effect sizes of a set of top associated SNPs on exposure and outcome to assess the genetic overlap (referred to as Pickrell-Cor). Note that Pickrell-Cor is similar to MR-Egger in the absence of pleiotropy despite that Pickrell et al. use a Bayesian approach to select SNPs. Pickrell et al. further exploit the asymmetry of correlation (the correlation computed from SNPs associated with the exposure is different from that computed from SNPs associated with the outcome) to infer causality by a maximum likelihood approach (referred to as Pickrell-ML).

In the previous version of our manuscript, we performed simulation to compare GSMR with MR-Egger and MR-IVW and showed that GSMR was more powerful than MR-Egger and MR-IVW especially when the number of independent instruments was large. This is because GSMR accounts for the sampling variance in both \hat{b}_{zx} and \hat{b}_{zy} where b_{zx} is the effect size of a SNP on exposure and b_{zy} is the effect size of the SNP on outcome whereas both MR-Egger and MR-IVW assume that b_{zx} is estimated without error.

In the revised manuscript, we have performed additional simulation to compare GSMR with the Pickrell methods (**Supplementary Fig. 3**). Result shows that the power of Pickrell-Cor is slightly

lower than that of MR-Egger because Pickrell-Cor does not account for the sampling variance in both \hat{b}_{zx} and \hat{b}_{zy} . The Pickrell-ML method tests for the asymmetry of the associations so that the power is limited (not comparable with the other methods).

Furthermore, GSMR has the advantage of accounting for correlations between SNPs in long-range linkage disequilibrium (LD) (Price et al. 2008 AJHG) although the generalized MR-IVW method (Burgess et al. 2012 Stat Med) also takes LD into account. Our new simulation shows the GSMR is more powerful than the generalized MR-IVW method again because the generalized MR-IVW approach assumes that b_{zx} is estimated without error (**Supplementary Fig. 3**).

In addition, we have shown by additional simulation that in the presence of pleiotropy the estimate of causal effect from GSMR (with HEIDI-outlier filtering) is unbiased whereas there is a small bias in the estimate from MR-Egger (**Supplementary Fig. 4**) that is thought to be free of confounding from pleiotropy.

We have further applied MR-Egger and the Pickrell methods to real data. Consistent with the results from simulation, the estimates from MR-Egger and the Pickrell approaches are less significant than those from GSMR (**Supplementary Table 7 and Supplementary Fig. 22**).

We have added the new analyses and results to the main text (pages 4 and 5), **Supplementary Figures 3 and 4, Supplementary Table 7 and Supplementary Note**.

Regarding to the evidence from randomised controlled trials (RCTs), there are a number of examples where the causal associations identified by our analysis have been confirmed by RCTs. These examples are BMI -> T2D (Look AHEAD Research Group 2010 Archives of Internal Medicine), LDL -> CAD (Baigent et al. 2005 Lancet), and SBP -> CAD (Collins et al. 1990 Lancet), which have been mentioned these studies in the main text (pages 7, 8 and 9). We have also indicated in **Supplementary Table 6** whether any of our significant results has been observed in a previous observational study or confirmed by a RCT.

A particular place where more analysis would help is in the analysis of HEIDI-outlier. They acknowledge it is not a "fail-safe" but the reader is left with little sense of how many pleiotropic effects (or what types/effect sizes) would escape detection. The mention a false positive rate of 0.01 given their lax p-value threshold, but more importantly, what is the false negative rate?

Re: We thank the reviewer for pointing out this.

The false positive rate of detecting horizontal pleiotropy is not important because even if a true causal signal is falsely detected as pleiotropy and eliminated from the analysis, it will only affect the power rather than the false positive rate or biasedness of the GSMR analysis. We chose a threshold of 0.01 because if there is no pleiotropy, only 1% of the valid instruments will be removed by chance, resulting in little loss of power. On the other hand, this is a relatively low p-value threshold for the heterogeneity test ensuring a relatively high power to detect pleiotropic outliers.

The power of detecting pleiotropy depends on the sample sizes and the deviation of a pleiotropic effect from the causal model. We have performed additional simulation (based on a causal model with pleiotropy) to quantify the power of detecting the pleiotropy effects (**Supplementary Fig. 4a**). There are certainly pleiotropic SNPs (especially those with small effect sizes) not detected by HEIDI-outlier. Nevertheless, those undetected pleiotropic effects do not seem to bias the GSMR estimate as demonstrated in our additional simulation (**Supplementary Fig. 4b**).

We have commented on this in the revised manuscript (page 5) and added the new simulation results in **Supplementary Figure 4**.

Even though the reverse causality tests generally revealed smaller effect sizes, they did find a reasonable number of hits even a strict FWER - this merits more explanation. Do these indicate false positives (more than the p-values would indicate, and thus potentially aren't calibrated), or are they suggesting these are actually true reverse effects (in addition to the forward effects?)?

Re: If the false positive rate at FWER is higher than expected, it suggests that p-value is not uniformly distributed under the null. We therefore performed additional simulation to calibrate the test-statistics for reverse-GSMR under the null hypothesis that there is a forward effect but no reverse effect (**Supplementary Note 4**). Result shows the reverse-GSMR p-value is indeed uniformly distributed under the null as demonstrated by the QQ-plot in **Supplementary Figure 20**, suggesting the false positive rate of reverse-GSMR analysis is well calibrated. We have added the new result to the revised manuscript (page 13).

We have further performed an MR-Egger regression analysis for the reverse effect. All of the estimated MR-Egger intercepts are very close to zero and none of them are significant, suggesting that the reverse effect is unlikely to be confounded from pleiotropy.

We therefore believe that the reverse effects are true in addition to the forward effects especially those in opposition direction with the forward effects (e.g. the bidirectional associations between BMI and T2D).

Other comments:

Can this be explained a little further (from Online Methods): "where there is a significant difference between b_{xy} estimated at an instrument i (i.e. $b_{xy}(i)$) and b_{xy} estimated at the SNP that shows the strongest association with exposure in the third quintile of the b^{\wedge}_{xy} distribution" -- why the third quintile?

Re: In the HEIDI-outlier test, we choose an instrument as a target and compare the estimate of b_{xy} at the target SNPs with those at the other instruments. We know that the power of detecting heterogeneity increases with the strength of association between the target SNP and exposure. However, we cannot simply choose the top exposure-associated SNP because sometimes when a SNP has an extremely strong effect on the exposure, it is also likely to be a pleiotropic outlier (e.g. the top LDL-associated SNP at the *APOE* locus shows a very strong pleiotropic effect on Alzheimer's disease, as shown in **Figure 4**). Therefore, to increase the robustness of the HEIDI-outlier test, we examine the distribution of \hat{b}_{xy} as a function of $-\log_{10}(\text{p-value})$ for \hat{b}_{zx} and choose the top exposure-associated SNP in the third quintile of the distribution to avoid choosing an extreme pleiotropic outlier as the target SNP.

We have clarified this in the revised manuscript (page 17).

The wording of this sentence is a bit convoluted: "We have previously shown that the power of MR could be greatly improved by a flexible analysis of summary-level GWAS data for exposure (e.g. risk factor) and outcome (e.g. disease) from two samples of large sample size (summary-data-based MR, SMR), and applied the SMR method to test if the effects of genetic variants on a phenotype are mediated by gene expression"

Re: We have revised the sentence as follows in page 3.

"We have previously shown that the power of an MR analysis could be greatly improved by exploiting GWAS summary data from two independent studies with large sample sizes, and have applied a Summary-data-based MR (SMR) approach to test if the effects of genetic variants on a phenotype are mediated by gene expression."

"estimate the effect of a risk factor on disease conditioning on other risk factors" - For a general audience should define "genetic value" the first time it's used.

Re: Done (page 18).

In the following sentences, they seem to contradict themselves. First saying there is likely mediation, then saying they can't really distinguish. I would tone down the first sentence: "suggesting that the marginal effects of HDL-c on diseases were likely to be mediated or driven by the covariate risk factors because of the complex bidirectional causative associations between HDL-c and the other risk factors as illustrated in Supplementary Fig. 9. It is difficult to distinguish the direction of mediation, i.e. whether HDL-c is a mediating (model I: other risk factors \rightarrow HDL-c \rightarrow disease) or driving (model II: HDL-c \rightarrow other risk factors \rightarrow disease) factor (Supplementary Fig. 13)."

Re: We have revised text as follows (page 11).

"However, all the effects became non-significant conditioning on the covariate risk factors (i.e. BMI, LDL-c, TG and SBP), suggesting that the marginal effects of HDL-c on the diseases are not independent of the covariates due to the complex bidirectional causative associations between HDL-c and the other risk factors as illustrated in **Supplementary Fig. 14**. It is difficult to distinguish whether the effects of HDL-c on the diseases are mediated or driven by the covariates (**Supplementary Fig. 17**) because of the complicated association network among risk factors and diseases (**Supplementary Fig. 15**)."

I would normally say that MR *uses* instrumental variables, not that MR *is* an instrumental variable. (from Introduction)

Re: Done (page 3).

Reviewer #2 (Remarks to the Author):

The authors proposed three summary-data-based methods in this manuscript: 1) to estimate and test the mediation effect of risk factor on disease status (b_{xy}) using multiple genetic variants as instruments, an extension of their previous SMR method 2) to detect outliers in the estimates of mediation effect which are likely caused by pleiotropy, i.e. not by mediation / causality 3) to estimate and test the mediation effect conditional on other risk factors. They applied their methods to multiple GWAS summary association data set to test for causal relationships between pairs of traits and diseases. Overall, I find the methods robust and useful addition to the community. However, more simulations and more rigorous analyses of real data are needed to demonstrate the robustness of their methods and validity of their conclusions.

Re: We thank the review for the positive remarks.

Major comments:

1) The authors developed GSMR to meta-analyze the estimates of effect of risk factor on disease status from multiple SNPs ($b_{zx,i}$) in a fixed-effect framework. This method models correlation between SNPs (LD) in the variance, and therefore is able to provide an unbiased and efficient estimate of the mediation effect. However, although the authors claim that their method can account for remaining LD after clumping, all of their simulations are based on independent SNPs – SNPs were drawn independently from the binomial. It would greatly strengthen the manuscript if

the author could perform simulations where the SNPs are not entirely independent from each other and then show that their method is still unbiased in the presence of LD.

Re: We thank the reviewer for the suggestion. We have conducted additional simulation based on SNPs in LD (**Supplementary Note 1.2.2**). The new simulation result shows that the estimate of b_{xy} from GSMR is unbiased in the presence of LD (**Supplementary Table 1**).

2) In the null simulation to test the unbiasedness of GSMR, the authors did not include any pleiotropic / direct effect of genetic effect on the disease status. For example, it could be the case that b_{xy} is zero, but multiple b_{zy} is not zero for both the exposure and the disease status at some of the SNPs. This could, in principle, results in bias estimate of b_{xy} , in the scenario where there is LD among the SNPs – the estimate of $b_{xy,i}$ at each SNPs are likely non-zero and highly correlated. It would greatly strengthen the manuscript if the authors could perform simulations where there is indeed pleiotropic / direct effect of genetic variants on the disease status. Also, although the authors demonstrated that the power of GSMR is not significantly affected by HEIDI-outlier filtering, an assessment of HEIDI-outlier in detecting the SNPs with pleiotropic effect would be very helpful.

Re: The power of detecting a pleiotropic SNP depends on the sample sizes of the GWAS data sets and the deviation of \hat{b}_{xy} estimated at the pleiotropic SNP from the causal model. We have performed simulation based on a model with pleiotropy to quantify the power of HEIDI-outlier to detect the pleiotropic effects (**Supplementary Fig. 4a**). There are certainly pleiotropic outliers (especially those with small effects) not detected by HEIDI-outlier. Nevertheless, these undetected pleiotropic effects do not seem to bias \hat{b}_{xy} from GSMR (**Supplementary Fig. 4b**), in contrast to a small bias in the estimate from Egger regression (MR-Egger) which is thought to be free of confounding from pleiotropy (Bowden et al. 2015 Int J Epidemiol).

The b_{xy} estimate by GSMR is also unbiased in the absence of causality but in the presence of pleiotropy (**Supplementary Table 2**).

We have added these new results in the revised manuscript (page 5).

3) The authors showed that their method is more powerful than Egger regression. However, this is not an entirely fair comparison, since Egger regression accounts for pleiotropy by incorporating an intercept term in the second regression step, effectively estimating two parameters instead of one as the case for GSMR. It's curious whether GSMR is still more powerful than Egger regression when there is indeed pleiotropy. Again, all the simulations performed in this study (as described in supplementary note) do not seem to make the assumption that there is indeed pleiotropy. An explanation of why GSMR is more powerful than Egger regression would also be helpful.

Re: Egger regression uses the intercept to account for pleiotropy while the GSMR analysis incorporates the HEIDI-outlier approach to detect and eliminate pleiotropic outliers. We have performed simulations to compare the methods in the presence of pleiotropy. The new simulation result shows that MR-Egger is even less powerful than GSMR in the presence of pleiotropy (**Supplementary Fig. 4c**).

4) In the absence of LD, the inverse-variance weighted approach to estimate the mediation effect should be equivalent to the GSMR approach (as shown by the equations at the top of page 23) and Egger regression. It's curious to me why GSMR has more power than IVW. An explanation would be helpful.

Re: IVW is the same as MR-Egger without intercept (Burgess et al. 2013 Genet Epidemiol and Bowden et al. 2015 Int J Epidemiol). Therefore, in the absence of pleiotropy these two methods

are equivalent. GSMR is different because GSMR takes the sampling variance (standard error squared) of both \hat{b}_{zx} and \hat{b}_{zy} whereas both MR-Egger and IVW assume b_{zx} is estimated without error. We have clarified this in the revised manuscript (page 5).

5) The authors analyzed many diseases that are case-control traits, which are often studied on ascertained samples, i.e. more cases are collected than in the general population to increase the chance of detecting the causal variants. The exposures, which are quantitative traits, on the other hand, are more likely studied on randomly collected (i.e. not ascertained) samples. The mismatch between the two types of samples will likely induce an overestimate of the mediation effect. The authors should elaborate more on how ascertainment could bias their causal inference.

Re: Because OR is free of the ascertainment bias in a case-control study, the effect (logOR) of a SNP on disease in the general population can be approximated by that from a case-control study. Therefore, GSMR can be applied to data with SNP effects on the risk factor from a population-based study and SNP effects on the disease from an ascertained case-control study, and the estimate of causative effect of the risk factor on the disease can be interpreted as that in the general population. We have clarified this in the revised manuscript (pages 6 and 17).

6) It's not clear whether the authors removed genetic variants in the HLA region before their analyses. If the author included SNPs in the HLA, an explanation would be helpful.

Re: The MHC region is often removed from the analysis in previous studies, mainly because of the complicated LD structure in this region. In this study, we did not remove this region because we use a set of near-independent SNPs as instruments after LD clumping. We have clarified this in the revised manuscript (page 20).

Minor comments:

1) The authors estimate $b_{\{xy\}}$ by meta-analyzing $b_{\{xy,i\}}$ of each SNP under a fixed-effect framework. Another approach is to assume the true $b_{\{xy\}}$ has a distribution, i.e. a random-effect framework. It's curious whether using a random-effect framework would be more appropriate as each SNP in the GWAS is likely tested under different sample sizes.

Re: Under a causal model, the expected value of \hat{b}_{xy} estimated at any of the instruments is constant. We therefore use a fixed effect. A random-effect model is more useful for analyses that allow heterogeneity (e.g. a meta-analysis of SNP effects across different populations).

2) The authors showed that reverse causation likely exist for BMI and T2D although the effect of T2D on BMI is much smaller. It's worth noting here that the sample sizes of GWASs on BMI and T2D are very different. Therefore, there can be biases in the number of GWS selected (e.g. number of GWAS for BMI likely larger than the number of GWS for T2D) in the estimation $b_{\{xy\}}$, which can in turn results in biases in $b_{\{xy\}}$ in both the forward and reverse direction. The authors should discuss this as potential bias of their analyses.

Re: To avoid an underpowered test, we limited the reverse-GSMR analysis to diseases that had more than 10 instruments. Indeed, some of the estimated reverse effects were small but highly significant (**Supplementary Table 15**). Therefore, it seems very unlikely that the large difference in the estimated effect sizes between the forward and reverse analyses is due to the lack of power in the reverse analysis. We further confirmed by simulation that the GSMR estimate of b_{xy} is unbiased irrespective of the sample size for the exposure (**Supplementary Fig. 21**). We have discussed this in the revised manuscript (page 13).

3) It would be helpful to provide an estimate of the remaining LD (e.g. the LD score) after applying

the clumping step in real data analyses.

Re: We have shown a distribution of LD scores of the instruments for each of the 7 exposures in **Supplementary Figure 8** after clumping with r^2 threshold = 0.05 and LD window size = 1Mb.

4) I would run the causal inference methods described in citation 16 to confirm / support the findings discovered in this manuscript.

Re: In the response to a comment from reviewer #1, we have explained the difference between the Pickrell methods (Pickrell 2016 Nat Genet) and the MR based methods (Burgess et al. 2013 Genet Epidemiol; Bowden et al. 2015 Int J Epidemiol; Burgess et al. 2016 Stat Med). The correlation method used in Pickrell et al. is similar to MR-Egger (Bowden et al. 2015 Int J Epidemiol) but slightly less powerful because it does not account for the sampling variance in both \hat{b}_{zx} and \hat{b}_{zy} . The Pickrell Maximum likelihood (Pickrell-ML) method tests the asymmetry of correlations in two directions, which is underpowered as demonstrated in our simulation (**Supplementary Fig. 3**). In fact, our results show that some of the causal effects can be bidirectional. Therefore, it is not always a good idea to use asymmetry to infer causality.

Nevertheless, we have included the Pickrell-ML methods in the analysis of real data. The estimates from Pickrell-ML are much less significant as those from GSMR or MR-Egger (**Supplementary Table 7**), consistent with the simulation result.

5) Adjusting for heritable covariates could induce bias in GWAS (see Aschard et al. AJHG 2015). Since most of the GWAS summary association data are likely adjusted for covariates, this could induce biases in estimating $b_{\{xy\}}$. Similarly, applying conditional analyses using summary data could result in false causative association as well.

Re: We thank the reviewer for pointing out this. The Aschard et al. (AJHG 2015) study quantifies the bias in the effect of SNP (g) on phenotype (Y) correcting for a covariate (C) if the C and Y are confounded by shared environmental effect (E) or uncorrelated genetic effect (G_g). If C and Y are standardized with mean 0 and variance 1, the bias is approximately $-\beta_C \rho_{CY}$ where β_C is the effect of g on C and ρ_{CY} is the correlation between C and Y . Our summary data based conditional analysis approach is free of this bias because we estimate ρ_{CY} from the GSMR approach. The expected value of ρ_{CY} estimated from GSMR is 0 if there is no direct effect of C on Y . This has been validated by simulation (**Supplementary Fig. 7**). We have commented on this in the revised manuscript (pages 5 and 19).

Reviewer #3 (Remarks to the Author):

Zhu et al. propose the method GSMR to estimate "causal" (with assumptions) relationships between traits from GWAS summary statistics. They extend their previous SMR statistic to estimate and test a Mendelian Randomization across multiple correlated instruments. They propose an outlier test to identify and remove SNPs that deviate from the overall MR trend, as well as a novel conditional analysis. Because the method only requires summary-level data, it was applied to a large number of risk factors and disease traits. They observe a causal effect of BMI on many traits (including, interestingly, overall disease count), as well as intriguing causal inferences between height, educational attainment and other traits.

Overall, the work aims to address a specific and important problem, proposes multiple methods, and describes interesting results. The manuscript is concise and easy to read. The conditional GSMR idea, in particular, is novel and informative. However, given that MR is now a mature research area with many competing methods, it is not clear what advance these methods offer over the most cutting-edge work, which is not cited or discussed very extensively. Moreover,

there are some issues with the SMR statistic and the simulation framework that left me unconvinced about the validity of the method. The results are interesting and of potential value to the field, but there are too many methodological gaps in the current form.

Re: We thank the review for the positive remarks.

Major Comments

* Recent MR methodology should be cited and compared to more thoroughly. Several methods to deal with correlated instrumental variables exist, of which weighted generalized linear regression (Burgess et al. Stat Med 2016 [PMID 26661904]) appears to be very similar in spirit to GSMR. What is the advance of this approach over the methods described in Burgess et al. and what is the relative performance of the methods? The approach of Pickrell et al. Nat Genet 2016 - perhaps the most recent high-profile MR paper - is cited but GSMR needs to be put in context to the causal inference in that paper (which analyzes many of the same traits) or compared by simulation. The same comments apply to the HEIDI outlier test: multiple summary-based sensitivity approaches are discussed in, for example, (Burgess et al. Epi 2017 [PMID 27749700]). How does HEIDI-outlier compare to those approaches?

Re: We have shown by additional simulation that GSMR is more powerful than the generalised MR-IVW method proposed by Burgess et al. (Stat Med 2016). The difference between the two approaches is that GSMR takes $\text{var}(\hat{b}_{zx})$ into account whereas MR-IVW or generalised MR-IVW assumes that b_{zx} is estimated without error (**Supplementary Fig. 3**).

Similarly the sensitivity analysis approaches described in Burgess et al. (2017 Epidemiology) also assume that b_{zx} is estimated without error and these methods do not account for LD between SNPs. We have commented on this in the revised manuscript (page 17). We have performed additional simulation based on a model with pleiotropy to quantify the power of HEIDI-outlier to detect the pleiotropic effects (**Supplementary Fig. 4a**). There are certainly pleiotropic outliers (especially those with small effects) not detected by HEIDI-outlier. Nevertheless, these undetected pleiotropic effects do not seem to bias the b_{xy} estimate by GSMR (**Supplementary Fig. 4b**), in contrast to a small bias in the estimate from MR-Egger (Bowden et al. 2015 Int J Epidemiol). The b_{xy} estimate by GSMR is also unbiased in the absence of causality but in the presence of pleiotropy (**Supplementary Table 2**).

We have also add the methods used in Pickrell et al. into comparison. The correlation method used in Pickrell et al. is similar to but slightly less powerful than MR-Egger (**Supplementary Fig. 3**) because the former does not account for the sampling variance in both \hat{b}_{zy} and \hat{b}_{zx} . The Pickrell Maximum likelihood (Pickrell-ML) method tests for the asymmetry of correlations in two directions, which is underpowered as demonstrated in our simulation (**Supplementary Fig. 3**). In fact, our results show that some of the causal effects can be bidirectional. Therefore, it is not always a good idea to use asymmetry to infer causality. Nevertheless, we have included the Pickrell-ML methods in the analysis of real data. The estimates from Pickrell-ML are much less significant as those from GSMR or MR-Egger (**Supplementary Table 7**), consistent with the simulation result.

We added these new results in the revised manuscript (pages 4 and 5).

* The derivation for $\text{var}(b_{xy})$ is approximate (pg.22) and is not well calibrated, casting doubt on the calibration of the subsequent statistics. Consider the following R snippet as a quick example:

```
set.seed(0)
chisqzy = rchisq(10e4,df=1,ncp=0)
chisqzx = rchisq(10e4,df=1,ncp=0)
```

```
smrstat = chisqzy * chisqzx / (chisqzy + chisqzx)
cat( mean(pchisq(smrstat,1) < 0.05) , '\n' )
```

In this null simulation 10% of statistics come up as significant at $P < 0.05$. Only after substantially increasing the non-centrality parameter for one of the traits does the empirical α start to approach 5%. This is a serious problem that the reader should be made aware of and I did not find any discussion of it in the manuscript. Especially since the actual causal estimate is identical to existing methods (Fig S16) so all of the power is coming from this variance estimator. This is the fundamental statistic used in all other tools so poor calibration has implications for all of the results. Please include a discussion of this issue and recommendation for how to avoid bias, and assure the reader that mis-calibration is not the reason GSMR performs better than existing methods.

Re: One of the basic assumptions of MR is that the instruments should be strongly associated with the exposure. Therefore, we cannot simply simulate “chisqzx” under the null without any ascertainment. In our analysis, we choose SNPs that are associated the exposure at $P < 5e-8$. If we modify the R code above according to the selection criterion used in GSMR (changes highlighted in yellow), then $\sim 5\%$ of statistics will be significant at $P < 0.05$, suggesting the false positive rate of GSMR is well calibrated under the null hypothesis that there is no effect of the exposure on outcome, consistent with our simulation result that there is no inflation in GSMR test-statistics under the null (**Supplementary Fig. 1**).

```
set.seed(0)
chisqzy = rchisq(10e4,df=1,ncp=0)
chisqzx = rchisq(10e4,df=1,ncp=29)
instruments = which(chisqzx > qchisq(5e-8, df = 1, lower.tail=F))
chisqzy = chisqzy[instruments]
chisqzx = chisqzx[instruments]
smrstat = chisqzy * chisqzx / (chisqzy + chisqzx)
cat( mean(pchisq(smrstat,1) < 0.05) , '\n' )
```

* All of the simulations in Supp Note 1 use independent SNPs drawn from a binomial distribution so the impact of LD is never actually evaluated. An LD correlation matrix is described in the Supplement (“In addition, we simulated 5,000 individuals in sample #3 (n_3) to calculate LD correlation matrix.”) but should only be contributing noise if the SNPs come from independent distributions. Please include thorough simulations with realistic population LD and reference panels (as in Supp Note 3 for example).

Re: We have performed additional simulation based on SNPs in LD. The result shows that the GSMR estimate is unbiased in the presence of LD (**Supplementary Table 1**).

* The GSMR method is presented as a generalization of MR that accounts for LD, and so I had expected it to be applied to most/all SNPs in the data. However, only genome-wide significant SNPs with highly strict LD pruning were used. It's not at all clear why these restrictions are necessary and they severely undercut the novelty of the approach since the remaining SNPs are nearly free of LD. Why throw away so much data? Given that this is such a key methodological point, I urge the authors to clearly explain and justify how SNPs should be selected for inclusion in the analysis to maximize power (this could be addressed using realistic LD simulations suggested in the previous comment).

Re: We only used genome-wide significant SNPs with stringent LD clumping criteria for the following two reasons.

1) Including SNPs in moderate to high LD often results in the **V** matrix being not invertible.

2) We used the near-independent genome-wide significant SNPs for the ease of directly comparing the results from GSMR with those from other methods that do not account for LD (e.g. MR-Egger).

We have conducted additional simulation based on SNPs in LD, and performed the GSMR analysis at different LD clumping thresholds. The result shows that the loss of power due to LD clumping is very small (**Supplementary Fig. 9**). We commented on this in the revised manuscript (page 6).

* There's also a broader question of what advantages this approach has over cross-trait LD-score regression (which was run on all pairs of traits anyway). The paper of Bulik-Sullivan et al. 2015 Nat Genet showed that cross-trait LDSC is asymptotically equivalent to the 2SLS Mendelian randomization estimate from the same set of variants, and confounding from pleiotropy affects both methods. So is there a clear advantage to using the GSMR statistics? Are there any instances where GSMR results are significantly different from cross-trait LDSC results? Is GSMR expected to have better power for certain disease architectures? I can see how the bi-directional GSMR approach gives you additional insights into causality, but is this better than running a sort of "bi-directional" LDSC on the top X% of SNPs from each trait?

Re: There are distinct features between LDSC and MR.

The bivariate LDSC method (Bulik-Sullivan et al. 2015 Nat Genet) aims to estimate the genetic correlation between two phenotypes. An estimate of genetic correlation, which takes a value between 0 and 1, can be interpreted as the estimated proportion of genetic effects shared in common between two traits. It accounts for both pleiotropic and causal effects, and does not test the direction of association.

The MR approaches aim to estimate and test for the causal effect of one phenotype on another. An MR estimate is interpreted as the estimated change in the outcome phenotype per unit increase of the exposure phenotype. An MR analysis tries to avoid the bias due to pleiotropic effects.

The bivariate LDSC method assumes a polygenic model for the traits and utilizes variability in LD scores across SNPs to estimate the genetic variance and covariance. Therefore, it has only been applied to estimate heritability and genetic correlation using all genome-wide SNPs. However, even if the LDSC method is applicable to a small subset of the SNPs, taking the top X% of SNPs from each trait for the LDSC method is not too dissimilar to the correlation and maximum likelihood methods used in the Pickrell et al. (2016 Nat Genet) study. Both methods are less powerful than GSMR as demonstrated by simulation (**Supplementary Fig. 3**). We commented on this in the revised **Supplementary Figure 3**.

Minor Comments

* For HEIDI-outlier please explain why the third quintile is used to define $b_{xy}(\text{top})$ and how the choice of quintile impacts the power and calibration of the method. Can the authors guarantee that using HEIDI-outlier to remove poor fitting SNPs and then running GSMR is always over-conservative?

Re: In the HEIDI-outlier test, we choose an instrument as a target and compare \hat{b}_{xy} at the target SNPs with those at the other instruments. The power of detecting heterogeneity increases with the strength of association between the target SNP and the exposure. However, we cannot simply choose the top exposure-associated SNP because sometimes when a SNP shows an extreme association signal with the exposure, it is also likely to be a pleiotropic outlier (e.g. the top LDL-associated SNP at the *APOE* locus shows a very strong pleiotropic effect on Alzheimer's disease as shown in **Figure 4**). Therefore, to increase the robustness of the HEIDI-outlier test, we examine

the distribution of \hat{b}_{xy} as a function of $-\log_{10}$ p-value for \hat{b}_{zx} and choose the top exposure-associated SNP in the third quintile of the \hat{b}_{xy} distribution to avoid choosing an extreme pleiotropic outlier as the target SNP. We have clarified this in the revised manuscript.

The power of detecting a pleiotropic SNP depends on the sample sizes of the GWAS data sets and the deviation of \hat{b}_{xy} estimated at the SNP from the causal model. We have performed additional simulation based on a causal model with pleiotropy to quantify the power of HEIDI-outlier to detect the pleiotropic effect (**Supplementary Fig. 4a**). There are certainly pleiotropic outliers (especially those of small effect) not detected by HEIDI-outlier. Nevertheless, these undetected pleiotropic effects do not seem to bias the GSMR estimate, in contrast to a small bias in the estimate from Egger regression (Bowden et al. 2015 Int J Epidemiol) which is thought to be free of confounding from pleiotropy (**Supplementary Fig. 4b**). The GSMR estimate of b_{xy} is also unbiased under a pleiotropic model without causal effect in the presence or absence of LD (**Supplementary Table 2**).

We have included these new results in the revised manuscript (pages 5 and 17), **Supplementary Table 2** and **Supplementary Figure 4**.

* The multi-trait conditioning requires h^2 and r^2 over the targeted SNPs, but uses LDSC estimates which are from all common variants. Is there a misspecification if the distribution of effect sizes is something like spike + slab and top hits have different h^2 and r^2 from the rest of the variants?

Re: The multi-trait conditional analysis (which is now called mtCOJO) is for all SNPs. Therefore, SNP-based heritability and genetic correlation should be estimated for all SNPs. We run the mtCOJO analysis for all SNPs and then re-select the instruments based on the conditional p-values. We have clarified this in the revised manuscript (page 19).

* The reverse GSMR statistics being less significant than forward GSMR is used as evidence of little pleiotropy: "Second, if the results were driven by pleiotropy, we would expect the estimates of b_{xy} from reverse GSMR comparable with those from GSMR, which is not what we observed". However, is this not strongly affected by differences in power between the two studies determining which instruments get selected? The GIANT and Edu traits are some of the largest GWAS in existence so it's not entirely surprising that associations in the reverse direction are weaker and may not be sufficient to rule out pleiotropy or reverse-causality.

Re: To avoid an underpowered test, we limited the reverse-GSMR analysis to diseases which had more than 10 instruments. Indeed, some of the estimated reverse effects were small but highly significant (**Supplementary Table 15**). Therefore, it seems very unlikely that the large difference in the estimated effect sizes between the forward and reverse analyses is due to the lack of power in the reverse analysis. We further confirmed by simulation that the GSMR estimate of b_{xy} is unbiased irrespective of the sample size for the exposure (**Supplementary Fig. 21**).

Nevertheless, we agree that there might be other sources of biases that could result in an underestimation of the reverse effect and have therefore removed this sentence.

Reviewers' comments:

Reviewer #2 (Remarks to the Author):

The authors have addressed most of my comments in a very satisfactory manner. And I only have one minor comment left.

Major comments:

- 1) This comment has been addressed effectively.
- 2) This comment has been addressed effectively.
- 3) This comment has been addressed effectively.
- 4) Makes sense.
- 5) OK.
- 6) I am a bit worried about the imputation quality of SNPs at MHC, which could result in inaccurate association scores during GWAS, which could then affect the result of GSMR.

Minor comments:

- 1) Makes sense.
- 2) This comment has been addressed effectively.
- 3) This comment has been addressed effectively.
- 4) This comment has been addressed effectively.
- 5) This comment has been addressed effectively.

Reviewer #3 (Remarks to the Author):

Thank you for the thorough discussion of the previous comments. The benefits of the GSMR approach are now much clearer and the additional simulations are very informative. I have two remaining presentation comments.

Comment 1

Thank you for performing additional simulations and a methods comparison. Is it correct that because the SNPs were simulated to be independent, the only gain from GSMR comes from modelling the $\text{var}(b_{zx})$? If so, I'm confused why MR-IVW and generalized MR-IVW perform differently since there is no correlation between markers. Please explain in the figure or indicate if not significantly different.

Comment 2

Thank you for the explanation of where GSMR is unbiased. Please add a discussion of this point to the main text, with a justification of which significance cutoff should be used to avoid bias. Currently the text on lines 100-104 states that there is no inflation in the GSMR test-static but this should be clarified to state that there is no inflation when only $\text{chisqz} > \text{the cutoff}$ are included and explicitly warn readers not to include non-significant variants.

Reviewers' comments

We thank all the two reviews for additional comments. We have responded to all the comments point-by-point as below in this document (in blue) and have highlighted all the relevant changes in the revised manuscript.

Reviewer #1:

N/A

Reviewer #2:

The authors have addressed most of my comments in a very satisfactory manner. And I only have one minor comment left.

Major comments:

- 1) This comment has been addressed effectively.
- 2) This comment has been addressed effectively.
- 3) This comment has been addressed effectively.
- 4) Makes sense.
- 5) OK.

6) I am a bit worried about the imputation quality of SNPs at MHC, which could result in inaccurate association scores during GWAS, which could then affect the result of GSMR.

Re: We have performed the GSMR analyses excluding the MHC SNPs. The results remain unchanged (**Supplementary Fig. 22**).

We have added the additional result in the revised manuscript (page 14 and **Supplementary Figure 22**).

Minor comments:

- 1) Makes sense.
- 2) This comment has been addressed effectively.
- 3) This comment has been addressed effectively.
- 4) This comment has been addressed effectively.
- 5) This comment has been addressed effectively.

Reviewer #3:

Thank you for the thorough discussion of the previous comments. The benefits of the GSMR approach are now much clearer and the additional simulations are very informative. I have two remaining presentation comments.

Comment 1

Thank you for performing additional simulations and a methods comparison. Is it correct that because the SNPs were simulated to be independent, the only gain from GSMR comes from modelling the $\text{var}(b_{zx})$? If so, I'm confused why MR-IVW and generalized MR-IVW perform differently since there is no correlation between markers. Please explain in the figure or indicate if not significantly different.

Re: We apologize for the confusion. We believe this comment refers to the results presented in **Supplementary Figure #**, where all the SNPs were simulated to be independent. Note that we focus on the results for independent SNPs because in practice we usually use near-independent

SNPs from a clumping analysis with a stringent LD r^2 threshold (e.g. 0.05), although GSMR accounts for remaining LD not removed by clumping (Supplementary Figure 1b and Supplementary Table 1). We have clarified this in our revised manuscript (Supplementary Figure #).

The difference between MR-IVW and generalized MR-IVW was due to the use of MR-Egger without intercept to compute the results for MR-IVW. Although the MR-Egger authors claim that MR-Egger without intercept is equivalent to MR-IVW (ref), there appears to be a small difference between the SE computed from the script provided by the MR-Egger authors with intercept 0 and that computed from our own script following the method described in MR-IVW (ref).

For consistency, we have re-computed the simulation results using the methods described in the original MR-IVW and generalized MR-IVW papers (refs). As expected, the results from the two methods are almost identical. There are subtle differences (not statistically significant) which are due to the use of SNP correlations (all chance correlations because the SNPs are independent) computed from a reference sample for generalized MR-IVW. We have clarified this in the revised manuscripts (Supplementary Figure #).

Comment 2

Thank you for the explanation of where GSMR is unbiased. Please add a discussion of this point to the main text, with a justification of which significance cutoff should be used to avoid bias. Currently the text on lines 100-104 states that there is no inflation in the GSMR test-static but this should be clarified to state that there is no inflation when only $\text{chisqzx} > \text{the cutoff}$ are included and explicitly warn readers not to include non-significant variants.

We thank the reviewer for the constructive suggestion. We have clarified in the revised version of our manuscript that the statement above is based on SNPs that are associated with the risk factor at $P < 5e-8$ (see page #). We have emphasized in both the Results and Discussion sections that only the SNPs that are strongly associated with the exposure (e.g. $P < 5e-8$) should be used as the instruments for GSMR and any other forms of MR analyses (see pages # and #).

REVIEWERS' COMMENTS:

Reviewer #2 (Remarks to the Author):

My comment has been satisfactorily addressed.

Reviewer #3 (Remarks to the Author):

The authors have fully addressed my comments. Thank you for the detailed responses.

REVIEWERS' COMMENTS:

Reviewer #2 (Remarks to the Author):

My comment has been satisfactorily addressed.

Re: We thank the reviewer for the comments on all the versions of our manuscript.

Reviewer #3 (Remarks to the Author):

The authors have fully addressed my comments. Thank you for the detailed responses.

Re: We thank the reviewer for the comments on all the versions of our manuscript.